# Dirac-fermion-assisted interfacial superconductivity in epitaxial topological-insulator/iron-chalcogenide heterostructures

Hemian Yi[1,8], Lun-Hui Hu[1,2,8], Yi-Fan Zhao[1], Ling-Jie Zhou[1], Zi-Jie Yan[1], Ruoxi Zhang[1], Wei Yuan[1], Zihao Wang[1], Ke Wang[3], Danielle Reifsnyder Hickey[3,4,5], Anthony R. Richardella[1], John Singleton[6], Laurel E. Winter[6], Xianxin Wu[7], Moses H. W. Chan[1], Nitin Samarth[1], Chao-Xing Liu[1]✉ & Cui-Zu Chang[1,3]✉

Over the last decade, the possibility of realizing topological superconductivity (TSC) has generated much excitement. TSC can be created in electronic systems where the topological and superconducting orders coexist, motivating the continued exploration of candidate material platforms to this end. Here, we use molecular beam epitaxy (MBE) to synthesize heterostructures that host emergent interfacial superconductivity when a non-superconducting anti-ferromagnet (FeTe) is interfaced with a topological insulator (TI) $(Bi, Sb)_2Te_3$. By performing in-vacuo angle-resolved photoemission spectroscopy (ARPES) and ex-situ electrical transport measurements, we find that the superconducting transition temperature and the upper critical magnetic field are suppressed when the chemical potential approaches the Dirac point. We provide evidence to show that the observed interfacial superconductivity and its chemical potential dependence is the result of the competition between the Ruderman-Kittel-Kasuya-Yosida-type ferromagnetic coupling mediated by Dirac surface states and antiferromagnetic exchange couplings that generate the bicollinear antiferromagnetic order in the FeTe layer.

Decoherence of a quantum state causes errors during the process of quantum computation and thus is the major obstacle to the realization of practical quantum computers. Majorana zero modes (MZMs), which obey non-Abelian statistics, are arguably the most promising anyon platform towards fault-tolerant quantum computing since they are expected to be inherently robust against local perturbation and decoherence effects. MZMs are expected to be bound to vortex cores in the presence of topological superconductivity (TSC) and appear as a zero-energy state inside the superconducting gap. The TSC phase can be created in hybrid structures where an electronic system with spin-split bands caused by strong spin-orbit coupling (SOC) is proximally coupled to a conventional $s$-wave superconductor[1–3]. This idea led to

[1]Department of Physics, The Pennsylvania State University, University Park, PA 16802, USA. [2]Department of Physics and Astronomy, The University of Tennessee, Knoxville, TN 37996, USA. [3]Materials Research Institute, The Pennsylvania State University, University Park, PA 16802, USA. [4]Department of Chemistry, The Pennsylvania State University, University Park, PA 16802, USA. [5]Department of Materials Science and Engineering, The Pennsylvania State University, University Park, PA 16802, USA. [6]National High Magnetic Field Laboratory, Los Alamos, NM 87544, USA. [7]CAS Key Laboratory of Theoretical Physics, Institute of Theoretical Physics, Chinese Academy of Sciences, Beijing 100190, China. [8]These authors contributed equally: Hemian Yi, Lun-Hui Hu. ✉e-mail: cxl56@psu.edu; cxc955@psu.edu

the fabrication of semiconductor nanowires covered with superconducting layers[4–7] and magnetic metallic wires on superconducting substrates[8,9]. So far, several experiments have reported evidence of MZM in 1D semiconducting nanowire/superconductor devices, such as unusual zero-bias conductance peak[4–6] and fractional AC Josephson effect[7]. However, there are serious concerns about the validity of such claims of MZMs in these devices[10–12]. These concerns provide new impetus to explore TSC in other systems.

Since topological insulators (TIs) possess strong SOC-induced inherent topological nontrivial band structures with inverted bulk conduction and valence bands, TIs provide a natural platform to pursue TSC. When a TI is coupled to a conventional $s$-wave superconductor through the proximity effect, TSC is expected to emerge[2]. Compared to the Rashba bands (i.e. the strong SOC-induced momentum-dependent band splitting) in 1D semiconducting nanowire systems[4,8,13,14], TI films have two advantages: (i) The helical Dirac surface states of TIs lift the spin degeneracy, thus bypassing the SOC-plus-Zeeman splitting control needed in 1D nanowire systems;[4] (ii) The bandgap of canonical TIs such as the Bi-chalcogenides is typically a few hundred meV[15,16] as compared to the magnetic field-induced Zeeman splitting of ~1 meV in 1D nanowire systems[4–6]. A large gap provides more flexibility for the realization of the TSC phase. Moreover, TI/superconductor heterostructures allow surface-sensitive probes such as angle-resolved photoemission spectroscopy (ARPES) and scanning tunneling microscopy and spectroscopy (STM/S) to directly probe the surface states. Such heterostructures also allow established transport techniques to be deployed for the exploration of the putative Majorana physics[2]. An ideal TI-based TSC platform would be comprised of an epitaxial heterostructure wherein both TI and superconductor layers are synthesized by molecular beam epitaxy (MBE), thus allowing precise engineering of the TI/superconductor interface and the band structure. However, there remains an important obstacle, specifically, the superconductivity in MBE-grown thin superconducting films usually disappears once a TI layer is grown on top, presumably due to the occurrence of charge transfer[17].

An alternative approach is to exploit the emergent interfacial superconductivity that arises in certain TI/non-superconducting material heterostructures that fulfill the two essential ingredients of TSC, i.e. topological and superconducting orders[2]. Recently, interfacial superconductivity has been observed in $Bi_2Te_3$/FeTe and $Sb_2Te_3$/FeTe heterostructures with superconducting temperatures of $T_c$ ~ 11.5 K and ~3.1 K, respectively[18,19]. In these two heterostructures, $Bi_2Te_3$ and $Sb_2Te_3$ are heavily electron- and hole-doped TI, respectively[15,20,21], while FeTe is an antiferromagnetic iron chalcogenide that is non-superconducting without element doping[22,23] and/or tensile stress[24]. Notably, a proximity effect-induced superconducting gap has been demonstrated on the top surface of the $Bi_2Te_3$/FeTe heterostructure,

setting the stage for the potential TSC phase[25]. Compared to bulk iron-based superconductors with a band topology such as $FeTe_{0.55}Se_{0.45}$ (ref. 26) and LiFeAs (ref. 27), the $(Bi,Sb)_2Te_3$/FeTe heterostructures provide the tunability of chemical potential across a single Dirac cone, without resorting to introducing disorder or applying strain in the sample to tune for the TSC phase.

In this work, we use MBE to synthesize $(Bi,Sb)_2Te_3$/FeTe heterostructures with an atomically sharp interface, in which the chemical potential of the ternary TI $(Bi,Sb)_2Te_3$ can be tuned by varying the Bi/Sb ratio[28,29]. By performing in vacuo ARPES and ex situ electrical transport measurements, we find that interfacial superconductivity appears in all $(Bi,Sb)_2Te_3$/FeTe heterostructures that we studied and we find both the superconducting transition temperature and the upper critical magnetic field are suppressed when the chemical potential approaches the Dirac point. This observation indicates the correlation between massless Dirac electrons of the TI layer and the interfacial superconductivity in $(Bi,Sb)_2Te_3$/FeTe heterostructures. We show evidence that the chemical potential dependence of interfacial superconductivity is a result of the competition between bicollinear antiferromagnetic order in the FeTe layer and the Ruderman-Kittel-Kasuya-Yosida (RKKY) interaction mediated by surface Dirac electrons of the TI layer[30].

All $(Bi_{1-x}Sb_x)_2Te_3$/FeTe ($0 \le x \le 1$) bilayer heterostructures are synthesized in an MBE chamber (ScientaOmicron), which is connected to an ARPES chamber. The MBE growth is monitored using in-situ reflection high-energy electron diffraction (RHEED) (Supplementary Fig. 1). The in vacuo ARPES measurements are carried out using a Scientia DA30L analyzer with unpolarized He-Iα light (~21.2 eV). Our MBE-grown $(Bi_{1-x}Sb_x)_2Te_3$/FeTe films are characterized by scanning transmission electron microscopy (STEM) (Fig. 1 and Supplementary Figs. 2–5), atomic force microscopy (Supplementary Fig. 6), and ARPES (Fig. 2 and Supplementary Figs. 7–11) measurements. The electrical transport studies are performed on mechanically scratched six-terminal Hall bar devices (Supplementary Fig. 12) with the magnetic field applied perpendicular to the film plane. More details about the MBE growth, sample characterization, and electrical transport measurements can be found in Methods.

## Results

Neither $(Bi,Sb)_2Te_3$ nor FeTe crystals by themselves are superconductors. The $(Bi,Sb)_2Te_3$ lattice is built from quintuple layer (QLs) containing two Bi layers and three Te layers[28,29], while the FeTe unit cell (UC) consists of one Fe layer and two Te layers (Fig. 1a). Although $(Bi,Sb)_2Te_3$ and FeTe have different lattice symmetry (hexagonal vs cubic, respectively), the van der Waals nature of both materials allows for epitaxial growth of $(Bi,Sb)_2Te_3$/FeTe heterostructures. To evaluate the changes of in-plane lattice constant in 8QL $(Bi_{1-x}Sb_x)_2Te_3$ layer on

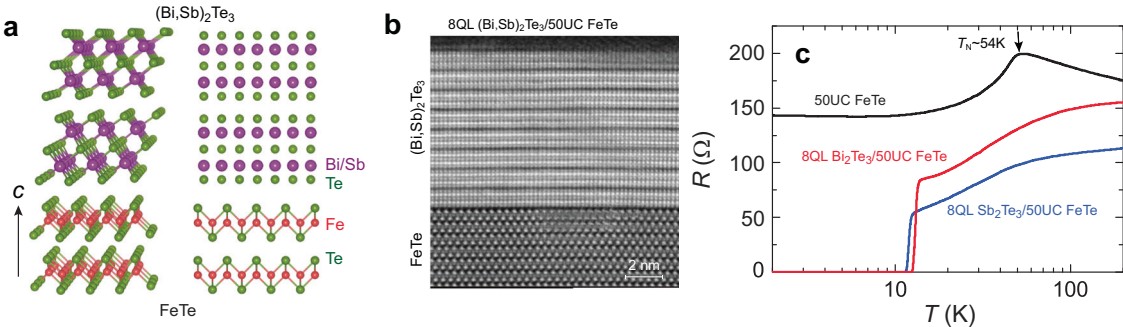

**Fig. 1 | Interfacial superconductivity in MBE-grown (Bi, Sb)₂Te₃/FeTe heterostructures. a** Schematic of the $(Bi, Sb)_2Te_3$/FeTe heterostructure. **b** Cross-sectional ADF-STEM image of the 8QL $(Bi, Sb)_2Te_3$/50UC FeTe grown on a heat-treated $SrTiO_3$ (100) substrate. **c** Temperature dependence of the sheet longitudinal resistance $R$ of 50UC FeTe (black), 8QL $Bi_2Te_3$/50UC FeTe (red), and 8QL $Sb_2Te_3$/50UC FeTe (blue) heterostructures. The hump feature in the black curve indicates that the *Néel* temperature $T_N$ of the 50UC FeTe film is ~54 K.

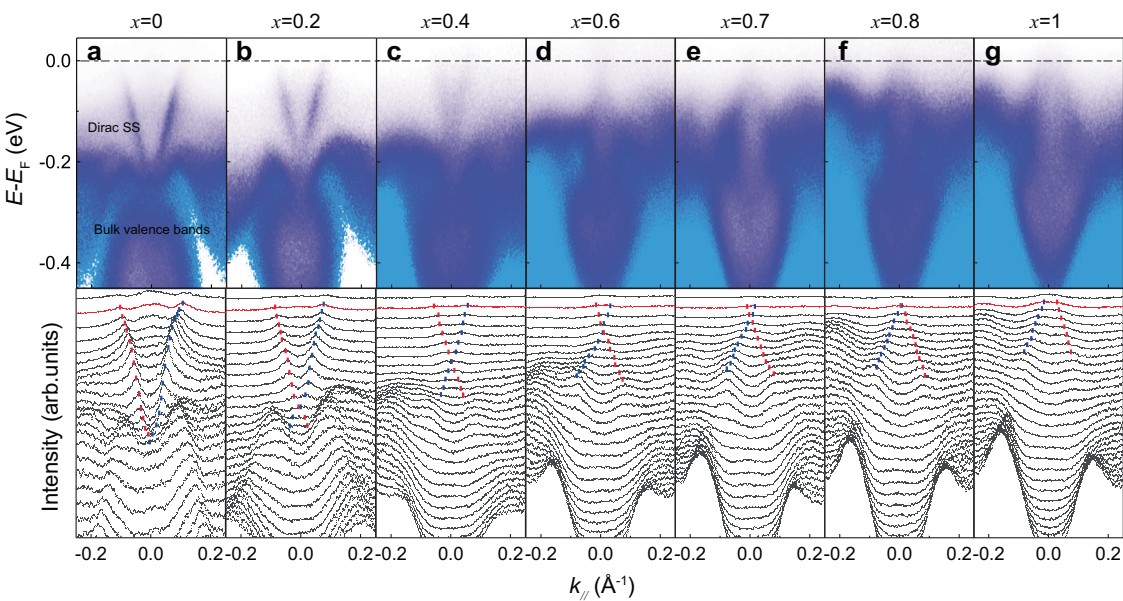

**Fig. 2 | Dirac surface states in 8 QL $(Bi_{1-x}Sb_x)_2Te_3$/50 UC FeTe heterostructures with different Sb concentrations $x$. a–g** Top: In vacuo ARPES band map of 8QL $(Bi_{1-x}Sb_x)_2Te_3$/50UC FeTe/SrTiO₃ heterostructures with $x = 0$ (**a**), $x = 0.2$ (**b**), $x = 0.4$ (**c**), $x = 0.6$ (**d**), $x = 0.7$, $x = 0.8$ (**f**), $x = 1$ (**g**). Bottom: The corresponding MDCs of the band maps in (**a–g**). SS, surface states. The dashed lines indicate the positions of the peaks in each momentum distribution curve. The linearly dispersed Dirac surface states are observed in all samples. The Dirac point moves from below to above the chemical potential by increasing $x$. All ARPES measurements are performed at room temperature.

50UC FeTe heterostructures, we extract the line intensity distribution curve from their RHEED patterns (Supplementary Fig. 1a). The spacing distance $d$ between the intensity peaks is inversely proportional to the lattice constant. We note that the $d$ value of the FeTe layer is close to that of the Bi₂Te₃ layer because the in-plane lattice constant ratio between FeTe and Bi₂Te₃ is $\sim \sqrt{3} : 2$ (ref. [31]). With increasing $x$, the value of $d$ expands, suggesting a gradual reduction of the lattice constant in the $(Bi_{1-x}Sb_x)_2Te_3$ layer (Supplementary Figs. 1g and 5). This will induce compressive stress if it exists in the FeTe layer. In Supplementary Fig. 5c, the $d$ value of FeTe is smaller compared to that of Sb₂Te₃, which is a result of the in-plane lattice mismatch between Sb₂Te₃ and FeTe layers. The monotonic decrease of the in-plane lattice constant indicates a van der Waals epitaxy growth mode in our $(Bi_{1-x}Sb_x)_2Te_3$/FeTe heterostructures. Moreover, we observe the twin domain feature with a rotation angle of ~30° in these heterostructures (Supplementary Figs. 5–7), further verifying an epitaxial growth of the heterostructures between two materials with hexagonal and cubic lattice structures[32]. Figure 1b shows the cross-sectional STEM image of an 8 QL (Bi,Sb)₂Te₃/50 UC FeTe heterostructure. We see the QL and trilayer atomic structures of (Bi,Sb)₂Te₃ and FeTe, respectively. Moreover, a highly ordered and sharp interface is seen between (Bi,Sb)₂Te₃ and FeTe even with different lattice structures. Supplementary Figs. 2–4 show the corresponding energy dispersive spectroscopy (EDS) maps of Fe, Bi, Sb, and Te, which confirm the automatically sharp interfaces in our (Bi,Sb)₂Te₃/FeTe heterostructures.

We next perform transport measurements on 50 UC FeTe, 8 QL Bi₂Te₃/50 UC FeTe, and 8 QL Sb₂Te₃/50 UC FeTe heterostructures. The 50 UC FeTe sample shows a semiconducting-to-metallic transition at $T \sim 54$ K. Such behavior in the R-T curve of FeTe has been associated with the paramagnetic-to-antiferromagnetic phase transition[19]. This magnetic ordering temperature is known as the *Néel* temperature $T_N$ (Fig. 1c). The spin fluctuation of the antiferromagnetic order has been proposed to be correlated to the superconductivity in Fe(Se,Te) single crystals[33]. For the 8 QL Bi₂Te₃/50 UC FeTe heterostructure (i.e. the $x = 0$ sample), the onset $T_{c,onset}$ and the zero resistance $T_{c,0}$ are ~13.6 K and ~12.5 K, respectively (Fig. 1c). For our Sb₂Te₃/FeTe heterostructure

(i.e. the $x = 1$ sample), $T_{c,onset}$ and $T_{c,0}$ are ~12.5 K and ~11.3 K, respectively (Fig. 1c). Compared to prior studies[18,19], the higher $T_c$ and much narrower superconducting transition windows observed here reveal that our MBE-grown Bi₂Te₃/FeTe and Sb₂Te₃/FeTe heterostructures are much more uniform and homogeneous.

To demonstrate the topological property of the $(Bi_{1-x}Sb_x)_2Te_3$ layer, we perform in vacuo ARPES measurements on a series of $(Bi_{1-x}Sb_x)_2Te_3$/FeTe heterostructures with different $x$ (Fig. 2 and Supplementary Figs. 9 and 10). The high quality of the FeTe layer is confirmed by observing a hole-type pocket near Γ point (Supplementary Fig. 8), consistent with prior studies[32]. In all $(Bi_{1-x}Sb_x)_2Te_3$/FeTe heterostructures with $0 \le x \le 1$, we observe linearly dispersed Dirac surface states at room temperature (Fig. 2 and Supplementary Fig. 9). We define the value of the Fermi momentum $k_F$ as the momentum at which the left branch of the Dirac surface states crosses the chemical potential. For the $x = 0$ sample, $k_F \sim -0.106$ Å⁻¹ and the Dirac point is buried in the bulk valence bands, ~245 meV below the chemical potential (Fig. 2a). A similar Fermi momentum $k_F$ is derived from ARPES band maps at low temperatures (Supplementary Fig. 11). With increasing $x$, the chemical potential moves downward and gradually approaches the Dirac point. Specifically, at $k_F \sim -0.082$ Å⁻¹, $-0.054$ Å⁻¹, $-0.025$ Å⁻¹, $-0.015$ Å⁻¹ for the $x = 0.2$, 0.4, 0.6, and 0.7 samples, respectively (Fig. 2b–e). For the $x = 0.8$ sample, the chemical potential almost crosses the Dirac point and $k_F \sim -0.005$ Å⁻¹ (Fig. 2f). With further increasing $x$, the chemical potential is below the Dirac point for the $x = 1$ sample and $k_F \sim 0.034$ Å⁻¹ (Fig. 2g). Therefore, our ARPES results show an electron- to hole-type crossover for the Dirac surface states in 8 QL $(Bi_{1-x}Sb_x)_2Te_3$/50 UC FeTe heterostructures. In addition, we find that with increasing $x$, one bulk valence band moves upwards, while the other one moves downwards (Supplementary Fig. 10).

To examine the role of the Dirac electrons in the development of the interfacial superconductivity, we perform electrical transport measurements on these 8 QL $(Bi_{1-x}Sb_x)_2Te_3$/50 UC FeTe heterostructures (Fig. 3). Figure 3a–h shows R-T curves of eight different $(Bi_{1-x}Sb_x)_2Te_3$/FeTe heterostructures with different $x$ under magnetic field $\mu_0H = 0$ T and $\mu_0H = 9$ T. As noted above, for the $x = 0$ sample, $T_{c,onset}$ and $T_{c,0}$ are ~13.6 K and ~12.5 K, respectively (Fig. 1c).

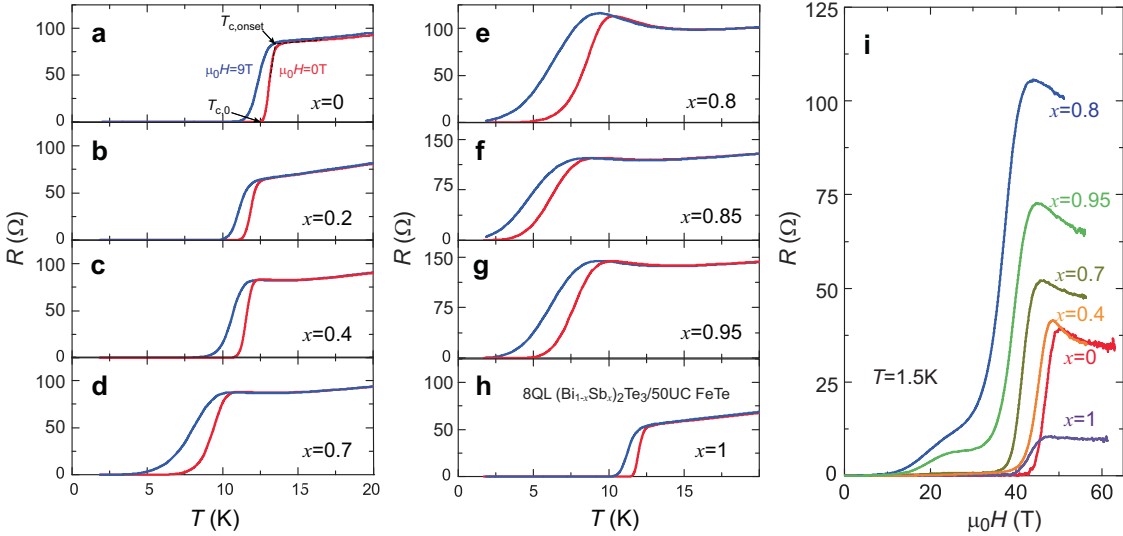

**Fig. 3 | Magnetotransport properties of 8QL $(Bi_{1-x}Sb_x)_2Te_3$/50UC FeTe heterostructures with interfacial superconductivity. a–h** Temperature dependence of the sheet longitudinal resistance $R$ measured at $\mu_0H = 0$ T (red) and $\mu_0H = 9$ T (blue) in 8QL $(Bi_{1-x}Sb_x)_2Te_3$/50UC FeTe/SrTiO$_3$ heterostructures with $x = 0$ (**a**), $x = 0.2$ (**b**), $x = 0.4$ (**c**), $x = 0.7$ (**d**), $x = 0.8$ (**e**), $x = 0.85$ (**f**), $x = 0.95$ (**g**), and $x = 1$ (**h**). **i** Magnetic field $\mu_0H$ dependence of $R$ of 8QL $(Bi_{1-x}Sb_x)_2Te_3$/50UC FeTe heterostructures measured at $T = 1.5$ K. The magnetic field $\mu_0H$ is applied along the out-of-plane direction.

With increasing $x$, both $T_{c,onset}$ and $T_{c,0}$ first decrease and then increase. We find $T_{c,onset}$ ~ 12.4 K, ~12.1 K, ~10.4 K, ~9.9 K, ~8.7 K, ~9.4 K, and ~12.5 K for the $x = 0.2$, 0.4, 0.7, 0.8, 0.85, 0.95, and 1.0 samples with corresponding $T_{c,0}$ ~ 10.9 K, ~10.5 K, ~5.5 K, ~3 K, ~1.7 K, ~4.1 K, and ~11.3 K. Moreover, we find that by applying a perpendicular magnetic field $\mu_0H = 9$ T, both $T_{c,onset}$ and $T_{c,0}$ decrease only slightly in all these samples. This observation implies a large upper perpendicular magnetic field ($\mu_0H_{c2,\perp}$) of the interfacial superconductivity in these $(Bi_{1-x}Sb_x)_2Te_3$/FeTe heterostructures.

To determine the values of $\mu_0H_{c2,\perp}$ of these $(Bi_{1-x}Sb_x)_2Te_3$/FeTe heterostructures, we perform electrical transport measurements under high perpendicular magnetic fields. Figure 3i shows $R$-$\mu_0H$ curves of six $(Bi_{1-x}Sb_x)_2Te_3$/FeTe heterostructures at $T = 1.5$ K. We define the value of $\mu_0H_{c2,\perp}$ as the magnetic field at which $R$ drops to half of the normal resistance. For the $x = 0$ sample, $\mu_0H_{c2,\perp}$ ~ 46.7 T, which is much larger than the value in a prior work[18]. With increasing $x$, the value of $\mu_0H_{c2,\perp}$ first decreases and then increases. Specifically, we find $\mu_0H_{c2,\perp}$ ~ 44.8 T, ~41.3 T, 36.3 T, ~38.8 T, and 43 T for the $x = 0.4$, 0.7, 0.8, 0.95, and 1.0 samples, respectively. Therefore, $\mu_0H_{c2,\perp}$ has a minimum near $x = 0.8$ (Fig. 3i). The sheet longitudinal resistance $R$ also shows a maximum value near $x = 0.8$, confirming our ARPES results that the chemical potential meets the Dirac point of the TI layer near $x = 0.8$ (Fig. 2). Supplementary Fig. 12 shows the more pronounced dip features of $\mu_0H_{c2,\perp}$ at 5 K, 7 K, and 9 K.

To reveal the link between Dirac surface states and the formation of the interfacial superconductivity in $(Bi_{1-x}Sb_x)_2Te_3$/FeTe heterostructures, we plot the values of $k_F$, $T_{c,onset}$, $T_{c,0}$, and $\mu_0H_{c2,\perp}$ as a function of the Sb concentration $x$ (Fig. 4a–c). For $x < 0.85$, the negative $k_F$ indicates the $n$-type Dirac fermion in the TI layer, $T_{c,onset}$, $T_{c,0}$, and $\mu_0H_{c2,\perp}$ decrease with increasing $x$. For $x > 0.85$, the positive $k_F$ indicates the $p$-type Dirac fermion in the TI layer, $T_{c,onset}$, $T_{c,0}$, and $\mu_0H_{c2,\perp}$ increase with increasing $x$. The values of $T_{c,onset}$, $T_{c,0}$, and $\mu_0H_{c2,\perp}$ show a minimum (Fig. 4b, c) when the chemical potential approaches the Dirac point, indicating that the Dirac electrons of the TI layer participate in the formation of the interfacial superconductivity. On the other hand, the formation of coherent Cooper pairs with condensed Dirac electrons is a prerequisite for the appearance of the TSC phase and Majorana physics in the $(Bi_{1-x}Sb_x)_2Te_3$/FeTe heterostructures.

## Discussion

In the following, we discuss the possible origins of the interfacial superconductivity and its chemical potential dependence in $(Bi_{1-x}Sb_x)_2Te_3$/FeTe heterostructures. The dip observed in the phase diagram (Fig. 4b) is likely a result of two main effects: (i) the charge transfer effect between $(Bi_{1-x}Sb_x)_2Te_3$ and FeTe layers, and (ii) the suppression of long-range antiferromagnetic order in the FeTe layer. First, both work functions in Bi$_2$Te$_3$ (-5.3 eV) and Sb$_2$Te$_3$ (-5.0 eV) are greater than that of FeTe (4.4 ~ 4.8 eV)[25,32,34]. This work function mismatch between $(Bi_{1-x}Sb_x)_2Te_3$ and FeTe is expected to cause a charge transfer, which may lead to the different energy levels for the Dirac points on the top and bottom [i.e. the $(Bi_{1-x}Sb_x)_2Te_3$/FeTe interface] surfaces of $(Bi_{1-x}Sb_x)_2Te_3$. As a consequence, the minimum value of $T_c$ is anticipated to be observed away from the Bi/Sb ratio at which the chemical potential crosses the Dirac point on the top surface of $(Bi_{1-x}Sb_x)_2Te_3$. In our experiments, the chemical potential crosses the Dirac point at $x = 0.8$ (Fig. 4a). However, the minimum values of both $T_{c,onset}$ and $T_{c,0}$ are observed at $x = 0.85$ (Fig. 4b). This observation further implies that the presence of band bending in the $(Bi_{1-x}Sb_x)_2Te_3$ layer. Prior studies[25,32] have claimed the occurrence of hole carrier transfer from Bi$_2$Te$_3$ to FeTe, which may give rise to interfacial superconductivity in the FeTe layer by screening out the strong Coulomb repulsion. However, in this doping effect hypothesis[35], the $T_c$ value of $(Bi_{1-x}Sb_x)_2Te_3$/FeTe heterostructures is likely to be enhanced or suppressed monotonically by tuning $x$ from 0 to 1. This assumption certainly cannot be the entire explanation in creating the interfacial superconductivity because it is inconsistent with the observed dip in our $T_c$ ~ $x$ phase diagram (Fig. 4b).

To understand the $T_c$ dip, we examine the second hypothesis that the origin of interfacial superconductivity in $(Bi_{1-x}Sb_x)_2Te_3$/FeTe heterostructures lies in the coupling with Dirac surface states that suppresses the long-range antiferromagnetic order of the FeTe layer. As noted above, our transport results indicate that the long-range antiferromagnetic order is weakened or even destroyed after the deposition of the $(Bi_{1-x}Sb_x)_2Te_3$ layer (Fig. 1c and Supplementary Fig. 14). Besides the charge transfer effect, we note that an RKKY magnetic exchange interaction in FeTe can be caused through its coupling to Dirac surface states near the interface[30,36], which could renormalize the spin-spin interactions and tune the magnetic ground state in the FeTe

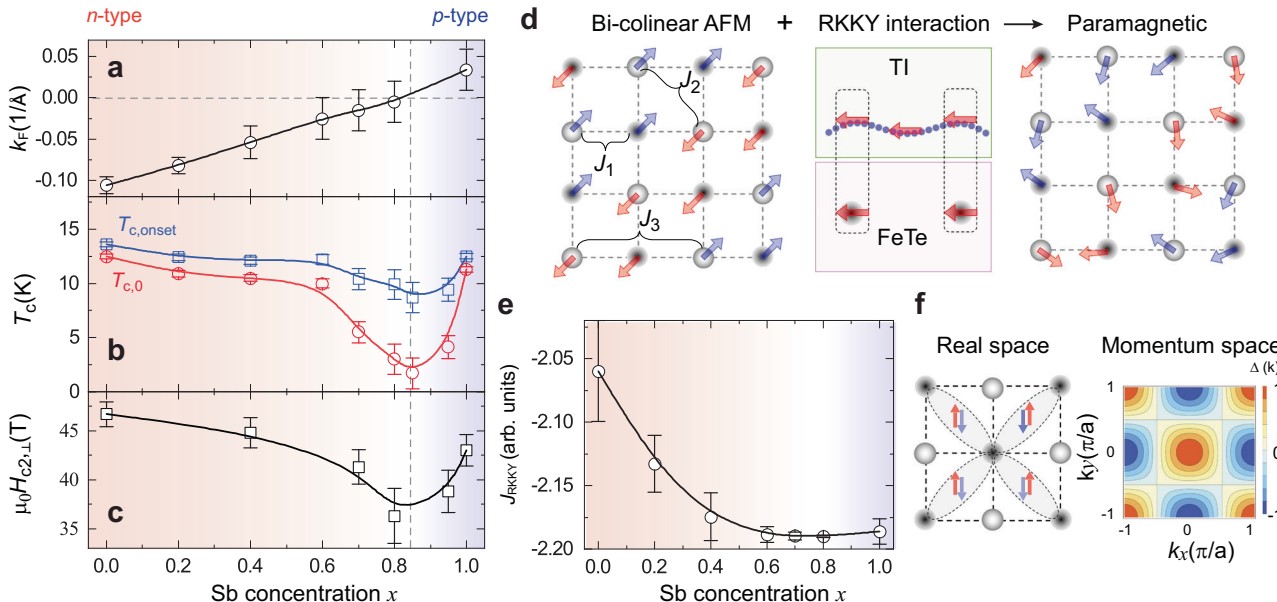

**Fig. 4 | Dirac-fermion-assisted interfacial superconductivity in $(Bi_{1-x}Sb_x)_2Te_3$/FeTe heterostructures. a–c** The Sb concentration $x$ dependence of the Fermi momentum $k_F$ (**a**), the onset of superconducting transition temperature $T_{c, onset}$ (blue squares) and the superconducting transition temperature with the zero resistance $T_{c,0}$ (red circles) (**b**), and the upper critical magnetic field $\mu_0 H_{c2,\perp}$ (**c**). The error bars in (**a**) are estimated from the width of the surface state at $k_F$. The error bars in (**b**) are estimated to be ~20% of ($T_{c, onset} - T_{c,0}$). The error bars in (**c**) are estimated to be ~40% of [$\mu_0 H_{c2,\perp} (R_{normal})$· $\mu_0 H_{c2,\perp} (R - 0.5R_{normal})$]. **d** A phenomenological interpretation of the Dirac-fermion-assisted interfacial superconductivity in $(Bi_{1-x}Sb_x)_2Te_3$/FeTe heterostructures. The left panel shows the bicollinear antiferromagnetic order of the FeTe layer. $J_i$ with $i = 1,2,3$ are spin-spin interactions. The middle panel shows the spin-exchange coupling between itinerant Dirac electrons and local spins of FeTe, resulting in the RKKY interaction between two magnetic moments. The right panel shows a possible renormalized spin model in FeTe with a suppressed antiferromagnetic order. **e** The Dirac fermion-induced RKKY interaction $J_{RKKY}(k_F R_{ij})$. Here $R_{ij}$ is the next nearest bond of irons, and it reaches its minimal for $k_F \to 0$ near $x \approx 0.85$. The error bars in (**e**) are estimated through the calculation of both $J_{RKKY}(k_F - \Delta k_F)$ and $J_{RKKY}(k_F + \Delta k_F)$, mirroring the error bars in (**a**). **f** The possible $s_\pm$-wave pairing symmetry in real and momentum spaces.

layer. For example, the antiferromagnetic order of FeTe can be described by the $J_1 - J_2 - J_3$ spin model, where $J_{1,2,3}$ are the magnetic exchange interactions (Fig. 4d) and are all positive (i.e. antiferromagnetic)[37]. Once the FeTe layer is coupled to the $(Bi_{1-x}Sb_x)_2Te_3$ layer, the local magnetic moments of Fe tend to be aligned to the spin direction of the Dirac surface states, leading to the RKKY interaction between different Fe magnetic moments. Prior studies[30,36,38,39] have shown that the Dirac surface states mediated RKKY interaction consists of Heisenberg-like, Ising-like, and Dzyaloshinskii-Moriya (DM)-like terms, all of which are not compatible with the bi-collinear antiferromagnetic order and can lead to strong magnetic fluctuations. Therefore, the RKKY interaction, in addition to the doping effect[25,32], may weaken or even destroy the bi-collinear antiferromagnetic order in the FeTe layer, so superconductivity is expected to emerge.

Next, we show that the RKKY interaction also explains the chemical potential dependence of $T_c$ observed in our experiments. As discussed in refs. 30,36, when the chemical potential approaches the Dirac point, the oscillation feature of RKKY interaction becomes weaker or even disappears as $k_F \to 0$ and Fermi wavelength $\lambda_F = \frac{1}{k_F} \to \infty$. Specifically, a correction to the spin–spin interaction in the FeTe layer can be found, $J_i' = J_i + J_{RKKY}(k_F a_i)$, from the RKKY interaction (Methods), where $J_{RKKY}$ is the RKKY-induced interaction at two sites with the distance $a_i$ ($i = 1,2,3$ label the nearest neighbor $a_1 = a_{FeTe}$, next nearest neighbor $a_2 = \sqrt{2}a_{FeTe}$, and the third nearest neighbor $a_3 = 2a_{FeTe}$) and has a negative sign (i.e. ferromagnetic type). Prior studies[40–43] have shown that that the dominant superconducting pairing is determined by the next nearest-neighbor exchange coupling $J_2'$ term, which gives rise to the spin-singlet $s_\pm$-wave pairing with the gap function form $\Delta(\vec{k}) = \cos k_x \cos k_y$ in iron chalcogenides (Fig. 4f and Methods)[44,45]. At the interface, $J_2 > 0$ from the intrinsic antiferromagnetic contribution in the FeTe layer should be dominant over

the correction of ferromagnetic $J_{RKKY} < 0$, which comes from the proximity from Dirac surface states to magnetic moments in FeTe. Therefore, we expect $J_2'$ constantly has a positive sign (antiferromagnetic). As the amplitude $|J_{RKKY}|$ increases first and then decreases with increasing Sb concentration $x$ (Fig. 4e), $J_2'$ is expected to first decrease and then increase accordingly. Consequently, the mean-field critical temperature $T_c \propto e^{-\frac{1}{3N_0 J_2'}}$ (Methods), where $N_0$ is the electron density of states, is expected to exhibit a non-monotonic behavior. We note that the parameters used in our theoretical calculations are extracted from our experiments. $|J_{RKKY}|$ is estimated to be ~1 meV for $k_F \to 0$, which is comparable to the intrinsic $J_2 \sim 10$ meV (Methods). When $J_2'$ reaches a minimum when the chemical potential is near the Dirac point ($k_F \to 0$), the superconducting pairing shows a dip in the $T_c - x$ phase diagram (Fig. 4b). Our experimental results thus suggest a mechanism of correlating the interfacial superconductivity with surface Dirac electrons in $(Bi_{1-x}Sb_x)_2Te_3$/FeTe heterostructures through the chemical potential tuning of spin-spin interaction across the interface.

Besides the above direct correction to short-range $J_i$ in the atomic scale, the long-range ferromagnetic component of the RKKY interaction may also arise in the scale of the superconducting coherence length. As the spin-singlet pairing is incompatible with ferromagnetic coupling, this long-range ferromagnetic component of the RKKY interaction may further weaken the superconductivity when the chemical potential approaches the Dirac point. We note that the DM component of the RKKY interaction also remains for the chemical potential near the Dirac point, which can support skyrmion spin texture[38]. A complete understanding of the influence of the complex RKKY interaction on the pairing symmetry and mechanism of the emergent interfacial superconductivity in $(Bi_{1-x}Sb_x)_2Te_3$/FeTe heterostructures needs further theoretical and experimental studies.

To summarize, we synthesize a series of $(Bi,Sb)_2Te_3/FeTe$ heterostructures and find that interfacial superconductivity appears in these heterostructures. By performing ARPES and electrical transport measurements, we find the superconducting temperature and the upper critical magnetic fields are suppressed when the chemical potential approaches the Dirac point. These observations reveal the correlation between the interfacial superconductivity and the Dirac surface states in the TI layer, indicating the Dirac electrons in the TI layer must play a role in the formation of the interfacial superconductivity. The dip in the $T_c \sim x$ phase diagram can be understood as a consequence of the complex competition between the RKKY interaction and antiferromagnetic exchange coupling. Our experiments provide evidence for Dirac-fermion-assisted interfacial superconductivity in $(Bi,Sb)_2Te_3/FeTe$ heterostructures and thus provide strong motivation for using this as a model material system for exploring Majorana physics and topological quantum computation. We also envision that extending $(Bi,Sb)_2Te_3/FeTe$ bilayers to multilayers could provide a rich platform for the pursuit of other emergent topological phenomena, including Weyl superconductors[46].

# Methods

## MBE growth

The $(Bi_{1-x}Sb_x)_2Te_3/FeTe$ heterostructures are synthesized in a commercial MBE system (ScientaOmicron) with a vacuum better than $2 \times 10^{-10}$ mbar. The insulating $SrTiO_3$ (100) substrates are first soaked in ~80 °C deionized water for 1.5 h and then put in a diluted hydrochloric acid solution (~4.5% w/w) for 1 h. These $SrTiO_3$ (100) substrates are then annealed at ~974 °C for 3 h in a tube furnace with flowing high-purity oxygen gas. Through the above heat treatments, the $SrTiO_3$(100) substrate surface becomes passivated and atomically flat, which becomes suitable for the MBE growth of the FeTe films. We next load the heat-treated $SrTiO_3$ (100) substrates into our MBE chamber and outgas them at 600 °C for ~1 h. High-purity Bi (99.9999%), Sb (99.9999%), Te (99.9999%), and Fe (99.995%) are evaporated from Knudsen effusion cells. The growth temperature is maintained at ~340 °C and ~210 °C for the MBE growth of the FeTe and $(Bi,Sb)_2Te_3$ layers, respectively. The growth rate is set to ~0.4 UC/min for the FeTe film, and ~0.2 QL/min for the $(Bi,Sb)_2Te_3$ layer, which is calibrated by measuring the thickness of the $(Bi,Sb)_2Te_3/FeTe$ bilayer heterostructures using both STEM and atomic force microscopy measurements. Finally, to avoid possible contamination, a ~10 nm Te layer is deposited at room temperature on top of the bilayer heterostructures before their removal from the MBE chamber for ex situ electrical transport measurements.

## ADF-STEM measurements

The aberration-corrected ADF-STEM measurements are performed on an FEI Titan[3] G2 STEM operating at an accelerating voltage of 300 kV, with a probe convergence angle of 25 ~ 30 mrad, a probe current of 70 ~ 100 pA, and ADF detector angles of 42 ~ 253 mrad. More STEM images and EDS maps of $(Bi,Sb)_2Te_3/FeTe$ heterostructures can be found in Supplementary Figs. 2–5.

## ARPES measurements

We perform *in-vacuo* ARPES measurements in a chamber with a base vacuum better than $5 \times 10^{-11}$ mbar. After finishing the MBE growth, the $(Bi,Sb)_2Te_3/FeTe$ bilayer heterostructures are transferred from the MBE chamber to the ARPES chamber. The energy analyzer DA30L (ScientaOmicron) is used. We use a helium-discharge lamp with a photon energy of ~21.2 eV. The energy and angle resolutions are set to ~10 meV and ~0.1°, respectively. We note that He $I_\alpha$ light with an energy of ~21 eV is sensitive to the sample surface[47].

## Electrical transport measurements

The $(Bi,Sb)_2Te_3/FeTe$ bilayer heterostructures are scratched into a Hall bar geometry using a computer-controlled probe station. The Hall bar device size is ~1 mm × 0.5 mm (Supplementary Fig. 12). We made the electrical contacts by pressing tiny indium spheres on the Hall bar, and the presence of the indium contacts does not influence the interface-induced superconducting in $(Bi_{1-x}Sb_x)_2Te_3/FeTe$ heterostructures (Supplementary Fig. 15). The electrical transport measurements are conducted using a Physical Property Measurement Systems (Quantum Design DynaCool 1.7 K, 9 T) and a capacitor-driven 65 T pulsed magnet at the National High Magnetic Field Laboratory (NHMFL), Los Alamos. The excitation current is ~1 μA for all $R$-$T$ measurements.

The Hall trace of the FeTe layer shows a nearly zero slope at low temperatures (Supplementary Figs. 16 and 17), which indicates a significantly higher carrier density in the FeTe layer. Consequently, the Hall trace of the $(Bi_{1-x}Sb_x)_2Te_3/FeTe$ heterostructure does not accurately reflect the carrier density (or the position of the chemical potential) in the $(Bi_{1-x}Sb_x)_2Te_3$ layer (Supplementary Fig. 17). For the $(Bi_{1-x}Sb_x)_2Te_3$ layer without the FeTe layer, a comprehensive analysis comparing $k_F$ derived from ARPES and Hall transport measurements has been carefully studied in our prior work[28]. Therefore, for the $(Bi_{1-x}Sb_x)_2Te_3/FeTe$ heterostructures used in this work, the $k_F$ value obtained from ARPES measurements is considered more reliable in reflecting the position of the chemical potential position of $(Bi_{1-x}Sb_x)_2Te_3$.

## X-ray diffraction reciprocal space mapping (RSM) measurements

We perform XRD reciprocal space mapping (RSM) measurements on $Bi_2Te_3/FeTe$ and $Sb_2Te_3/FeTe$ heterostructures, respectively (Supplementary Fig. 18). RSMs are collected on a 320 mm radius Panalytical X'Pert[3] MRD four circle X-ray diffractometer equipped with a line source [Cu $K_{\alpha1\text{-}2}$ (1.540598/1.544426 Å)] X-ray tube operating at ~45 kV and ~40 mA. The incident beam path includes a 2×Ge(220) asymmetric hybrid $K_{\alpha1}$ monochromator and a 1/4° divergence slit. A PIXcel3D 1×1 detector operating in frame-based mode is used to measure the RSMs. The lower and upper levels of the pulse height distribution are set at ~4.02 and ~11.27 keV, respectively. The heterostructures are aligned using the $SrTiO_3$(002) reflection and the azimuthal angle for off-axis peaks is aligned using the φ scans. The in-plane lattice constant is determined by measuring the off-axis peaks in the symmetric geometry after tilting the sample in χ because the reflections available for asymmetric scans had very low intensities. The 2θ positions for reflections are measured by fitting a Lorentzian function to the data.

For the 8 QL $Bi_2Te_3$/50 UC FeTe heterostructure, the out-of-plane $Bi_2Te_3$(0 0 15) and off-axis (0 1 5) reflections are measured (Supplementary Fig. 18a and b), which are used to determine the in-plane and out-of-plane lattice constants. The calculated strain in the $Bi_2Te_3$ layer is ~1% in-plane tensile strain, with no significant strain observed out of the plane. For the 8 QL $Bi_2Te_3$/50 UC FeTe heterostructure, a light strain is observed in the in-plane or out-of-plane directions (~0.3%) (Supplementary Fig. 18d and e). The presence of a small strain in the $(Bi_{1-x}Sb_x)_2Te_3$ layer grown on the FeTe layer implies no changes in the topological properties of $(Bi_{1-x}Sb_x)_2Te_3$ (refs. 48,49). Moreover, the twin domains with a 30° rotation are further verified by φ scans of $(Bi_{1-x}Sb_x)_2Te_3$/50 UC FeTe heterostructures (Supplementary Fig. 18c and f), which are consistent with our RHEED, atomic force microscopy, and ARPES measurements (Supplementary Figs. 1, 6, and 7).

## Surface Dirac electrons mediated RKKY interaction

We investigate the RKKY interaction within adjacent magnetic moments on the TI surface (Supplementary Fig. 19). The two magnetic moments are labeled by $\vec{S}_1$ and $\vec{S}_2$ (Supplementary Fig. 19a). The TI surface here can serve as the interface that is coupled with the FeTe layer. The magnetic moments likely have their origin in the extra iron impurities on the surface of the FeTe layer near the TI/FeTe interface. To simplify our theoretical model, we only consider the Dirac surface states of the TI layer (Supplementary Fig. 19b). This simplification is

scientifically sound because the chemical potential is located within the bulk band gap as the Sb concentration $x$ varies from 0 to 1. Moreover, unlike the oscillating RKKY interaction in a conventional metal, the RKKY of the TI/FeTe heterostructure shows a relatively strong ferromagnetic-type RKKY interaction, primarily induced by a small $k_F$. Here the $k_F$ value of the surface Dirac electrons is much less than that of the TI bulk states.

The two magnetic moments $\vec{S}_1$ and $\vec{S}_2$ are strongly coupled via the itinerant Dirac electrons on the TI surface (Supplementary Fig. 19b). The Hamiltonian for this system is $H = H_{surf} + H_{int}$, where $H_{surf} = A(k_x \sigma_y - k_y \sigma_x)$ and $H_{int} = -J_{eff}(\vec{S}_1 \cdot \vec{\sigma}_c(\vec{R}_1) + \vec{S}_2 \cdot \vec{\sigma}_c(\vec{R}_2))$. Here $A$ is the Fermi velocity of Dirac electrons, $(k_x, k_y)$ is the momentum, $J_{eff}$ is the effective spin-spin interaction strength, $\vec{R}_{1,2}$ are the location of magnetic moments $\vec{S}_{1,2}$, and $\vec{\sigma}_c$ is the spin operator for the Dirac electrons in real space.

The RKKY interaction between $\vec{S}_1$ and $\vec{S}_2$ is given by $J_{RKKY} = \frac{J_{eff}^2 V_0^2}{A R_{ij}^3} \times \Phi_{zz}(k_F R_{ij})$, where $V_0$ is the in-plane area of the TI unit cell and $R_{ij}$ is the in-plane spatial distance between the two magnetic moments[30]. Note that the constant coefficient $\frac{J_{eff}^2 V_0^2}{A R_{ij}^3}$ provides us the energy unit. The dimensionless integral part $\Phi_{zz}(z)$, as a function of dimensionless parameter $z = k_F R_{ij}$, is the addition of intra-band and inter-band contributions. $\Phi_{zz}(z) = F_{intra}(z) + F_{inter}(z)$, where $F_{intra}(z) = \int_0^z \frac{x dx}{2\pi} \int_z^\Lambda \frac{x' dx'}{2\pi} \frac{1}{x - x' + i\Gamma} [J_0(x)J_0(x') - J_1(x)J_1(x')]$ and $F_{inter} = \int_0^\Lambda \frac{x dx}{2\pi} \int_z^\Lambda \frac{x' dx'}{2\pi} \frac{1}{-x - x' + i\Gamma} [J_0(x)J_0(x') + J_1(x)J_1(x')]$ with $\Lambda$ is the cut-off. In our calculations, we choose $\Lambda = 120$ which is sufficiently large because of the maximal $k_F R_{ij} \sim 1$. Here $i\Gamma$ is the phenomenological imaginary self-energy correction that describes the disorder broadening. The real part of self-energy provides corrections to the energy of Dirac points and the velocity of the Dirac surface states[50]. The RKKY interaction as a function of $z$ at different self-energies is shown in Supplementary Fig. 19c. We find that $\Phi_{zz}(z)$ curves become monotonic when $\Gamma \geq 0.3$. In the region with a larger $\Gamma$, the RKKY interaction is constantly ferromagnetic at a small $z$. For a selected $R_{ij} = \sqrt{2} a_{FeTe}$ for the next nearest neighbor sites, where $a_{FeTe}$ is the lattice constant of FeTe, we expect the RKKY interaction can reach a maximal value for $k_F \sim 0$ (Supplementary Fig. 19d). We plot $\frac{J_{eff}^2 V_0^2}{A R_{ij}^3} \times \Phi_{zz}(k_F R_{ij})$ as a function of $R_{ij}$ at different $k_F = 1.5, 1.0,$ and $0.5$. When $R_{ij}$ is small, the ferromagnetic interaction strength is large for a small $k_F$. Only for a relatively large $k_F$, the intra-band scattering contribution $F_{intra}$ dominates the RKKY interaction strength. Therefore, the typical RKKY oscillations are observed in real space (the black curve in Supplementary Fig. 19d), similar to the RKKY behavior in a conventional metal. For a small $k_F$, the inter-band scattering contribution $F_{inter}$ dominates, which shows a strong ferromagnetic type RKKY interaction (the green curve in Supplementary Fig. 19d).

We employ the above method to calculate the strength of RKKY interaction in the (Bi,Sb)$_2$Te$_3$/FeTe heterostructure. $R_{ij} = \sqrt{2} a_{FeTe}$ and self-energy $\Gamma = 0.6$ are used. The values of $k_F$ are extracted from our ARPES data (Fig. 2). The estimated RKKY strength in (Bi,Sb)$_2$Te$_3$/FeTe heterostructures is shown in Fig. 4e. We also consider the weak $x$ dependence of $A$ that leads to error bars. $J_{RKKY} < 0$ is negative, revealing a ferromagnetic-type RKKY interaction. When $k_F$ is near the Dirac point, the inter-band contribution dominates, and a larger RKKY interaction $J_{RKKY}$ is achieved (Supplementary Fig. 19). Note that similar results are expected for smaller values of $R_{ij}$. Below we list the parameters used in our theoretical calculations. For different Sb concentrations $x$, the values of $k_F$ are extracted from our ARPES band maps in Fig. 2 and are plotted in Fig. 4a. Based on our ARPES band maps, we obtained the Fermi velocity $A \approx 3.5 eV \cdot Å$. The area $V_0 = \frac{3\sqrt{3}}{2} a_0^2 \approx 49.84 Å^2$ and the distance $R_{ij} \approx 5.34 Å$. In FeTe, the value of

$J_{eff}$ has been determined to be ~50 meV (Ref. 51). Therefore, we assume a comparable energy scale range of [10,100] meV for $J_{eff}$ and estimate the ferromagnetic-type RKKY strength $J_{RKKY}$ within the range of $-[0.02, 1.9] meV$ at $k_F \to 0$. This value is in the order of $1 meV$, which can greatly reduce the antiferromagnetic coupling in FeTe, whose energy scale can be estimated from the critical temperature of the bi-colinear AFM order ($T_N \sim 68$ K)[52,53]. Therefore, it can significantly affect $T_c$ of the interface-induced superconductivity in our (Bi,Sb)$_2$Te$_3$/FeTe heterostructures.

As discussed in the main text, the ferromagnetic type RKKY interaction competes with the superconducting order near the TI/FeTe interface, which can lower $T_c$. Our theoretical calculations point out a possible macroscopic model to understand the superconducting $T_c$ dip near the Dirac point of the TI layer. The $J_2$ term for the anti-ferromagnetic interaction between next-nearest Fe atoms is renormalized as $J_2' = J_2 + J_{RKKY}(k_F R_{ij})$ and thus alters by varying $k_F$. Though the sign of $J_{RKKY}$ is negative, which make $J_2'$ smaller than $J_2$. But $J_2'$ is still assumed to be positive, which enables tunable superconductivity (see "**Spin-singlet $s_\pm$ wave superconductivity**" below). In the context of the standard mean-field theory, the decomposition of the $J_2'$ term generally leads to the $s_\pm$ pairing (Fig. 4f)[40]. The smaller $J_2'$ indicates the lowering of $T_c$. Therefore, our experimental and theoretical studies suggest that the Dirac electrons participate in the formation of the interfacial superconductivity in (Bi,Sb)$_2$Te$_3$/FeTe heterostructures.

## Spin-singlet $s_\pm$ wave superconductivity

We employ the mean-field theory to study the $s$-wave pairing symmetry in iron-based superconductors, which is induced by the $J_2$ term[40–43]. By solving the linearized gap equation, we will derive the corresponding mean-field $T_c$. The full Hamiltonian consists of the non-interaction part $H_0(\vec{k})$ and the interactions $H_J$, where $H_0(\vec{k}) = \sum_n \sum_k \sum_\sigma \epsilon_{n,\vec{k}} c_{n,\vec{k},\sigma}^\dagger c_{n,\vec{k},\sigma}$ with $n$ representing the band index, $\sigma = \{\uparrow, \downarrow\}$ for the spin degrees of freedom. $\epsilon_{n,\vec{k}}$ is the $n$-th band dispersion. Spin-orbit coupling is ignored for simplicity. The interaction Hamiltonian describes the intra-band spin–spin interaction, $H_{J_2} = J_2 \sum_{n, \ll i,j \gg} \vec{S}_{n,i} \cdot \vec{S}_{n,j}$, where we neglect other interactions because the $J_2$ term is the dominant channel[54].

Next, we consider the mean-field decomposition for the $J_2$ term. With the fermion representation for the spin operators, $S_{n,i}^+ = S_{n,i}^x + iS_{n,i}^y = c_{n,i,\uparrow}^\dagger c_{n,i,\downarrow}$, $S_{n,i}^- = S_{n,i}^x - iS_{n,i}^y = c_{n,i,\downarrow}^\dagger c_{n,i,\uparrow}$ and $S_{n,i}^z = \frac{1}{2}(c_{n,i,\uparrow}^\dagger c_{n,i,\uparrow} - c_{n,i,\downarrow}^\dagger c_{n,i,\downarrow})$, we obtain $\vec{S}_{n,i} \cdot \vec{S}_{n,j} = -\frac{3}{4} b_{n,ij}^\dagger b_{n,ij} + ...$ where $b_{ij}$ is the leading spin-singlet Cooper pair operator $b_{n,ij} = \frac{1}{\sqrt{2}}(c_{n,i,\uparrow} c_{n,j,\downarrow} - c_{n,i,\downarrow} c_{n,j,\uparrow})$ and "..." represents other channels. For the leading channel, $H_{J_2} = -\frac{3}{4} J_2 \sum_{n, \ll i,j \gg} b_{n,ij}^\dagger b_{n,ij}$, we can perform a straightforward transformation from real space to momentum space, and obtain $H_{J_2} = -\frac{1}{N} \sum_{\vec{k}, \vec{k}'} V_{\vec{k}, \vec{k}'} c_{n,\vec{k},\uparrow}^\dagger c_{n,-\vec{k},\downarrow}^\dagger c_{n,-\vec{k}',\uparrow} c_{n,\vec{k}',\downarrow}$ with $V_{\vec{k}, \vec{k}'} = 3J_2 \cos k_x \cos k_y \cos k_x' \cos k_y'$ and $N$ the number of lattice sites. As the spin model has an antiferromagnetic coupling, i.e., $J_2 > 0$, the corresponding electron-electron interaction $H_{J_2}$ is attractive and thus allows for developing superconductivity.

After transforming the spin-spin interaction Hamiltonian to the electron-electron interaction Hamiltonian, we can perform the mean field decomposition to derive the linearized gap equation $3J_2\chi_0(T) = 1$, where $\chi_0(T)$ is the superconductivity susceptibility as defined below, to compute the mean-field superconducting critical temperature $T_c$. The superconductivity susceptibility is derived as $\chi_0(T) = -\frac{1}{k_B T} \sum_{\vec{k}} \sum_{\omega_n} \text{Tr}[(\cos k_x \cos k_y)^2 G_e(\vec{k}, i\omega_n) G_h(-\vec{k}, i\omega_n)]$, where $G_e(\vec{k}, i\omega_n) = \frac{1}{i\omega_n - \epsilon_{\vec{k}}}$, $G_h(\vec{k}, i\omega_n) = \frac{1}{i\omega_n + \epsilon_{\vec{k}}}$, and $\omega_n = (2n+1)\pi k_B T$ is the Matsubara frequency of fermions. The form factor $\cos k_x \cos k_y$ in the above $\chi_0(T)$ comes from the attractive interaction $V_{\vec{k}, \vec{k}'}$ in the

Hamiltonian $H_{J_2}$. With a simple parabolic dispersion $\epsilon_{\vec{k}} = \frac{k^2}{2m} - \mu$ for electron pockets, we can obtain $\chi_0(T) = \frac{N_0}{4} \log(\frac{2e^{\gamma}\omega_D}{\pi k_B T})$ with $N_0$ the electron density on the Fermi energy, $\gamma = 0.57721$ the Euler-Mascheroni constant and $\omega_D$ is the large energy cutoff. Therefore, the linearized gap equation gives the mean-field $T_c = \frac{2e^{\gamma}\omega_D}{\pi} e^{-\frac{4}{3N_0 J_2}}$. As we expect, $T_c$ decreases exponentially as $J_2$ decreases (assuming $J_2 > 0$). As we discussed in the main text, the decrease of $J_2$ is caused by the RKKY interaction mediated by the surface Dirac electrons, which explains the $T_c$ dip observed in our experiments.

## Data availability

The datasets generated during and/or analyzed during this study are available from the corresponding author upon request.

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

## Acknowledgements

We are grateful to Yongtao Cui, Weida Wu, Xiaodong Xu, and Kaijie Yang for their helpful discussions. This project is primarily supported by the DOE grant (DE-SC0023113) (C.-Z.C.), including the MBE growth and electrical transport measurements. The sample characterization is partially supported by the NSF-CAREER award (DMR-1847811) (C.-Z.C.) and the Penn State MRSEC for Nanoscale Science (DMR-2011839) (C.-Z.C.). The MBE growth and the ARPES measurements are performed in the NSF-supported 2DCC MIP facility (DMR-2039351) (C.-Z.C.). The theoretical calculations are partially supported by the NSF grant (DMR-2241327) (C.-X.L.). C.Z.C. acknowledges the support from the Gordon and Betty Moore Foundation's EPiQS Initiative (GBMF9063 to C.-Z.C). D.R.H. acknowledges the startup funds from Penn State. Work done at NHMFL is supported by NSF (DMR-1644779 and DMR-2128556) (J.S. and L.E.W.) and the State of Florida.

## Author contributions

C.-Z.C. conceived and designed the experiment. H.Y., Y.-F.Z., L.-J. Z., Z.-J.Y., R.Z., W.Y., Z.W., A.R.R., M.H.W.C., N.S., and C.-Z.C. performed the MBE growth, ARPES, and PPMS transport measurements. K.W. and D.R.H. performed the TEM measurements. J.S. and L.E.W. performed the transport measurements at NHMFL. L.H., X.W., and C.-X.L. provided theoretical support. H.Y., L.H., C.-X.L., and C-Z.C. analyzed the data and wrote the manuscript with input from all authors.

## Competing interests

The authors declare no competing interests.
