## [Peer Review File · Nature Communications]

Dirac-Fermion-Assisted Interfacial Superconductivity in Epitaxial Topological Insulator/Iron Chalcogenide HeterostructuresREVIEWER COMMENTS

Reviewer #1 (Remarks to the Author):

H. Yi, et al grow TI/Fe chalcogenide epitaxial heterostructures to study interfacial superconductivity and the role of Dirac fermions. The central claim of the paper is that by changing the composition of the TI the Fermi level moves through the Dirac node which leads to a decrease in T_c on either side of the Dirac node suggesting Dirac fermions promote the interfacial superconductivity. This result leads to new platforms to study topological superconductivity, especially given the consensus around super-semi Majoranas is not agreed upon. The authors present a rather clear picture of the data, though some of it could be cleaned up and made more convincing. Below is a list of points that should be addressed.

Main Points:

- The experimental evidence seems to support the statement that the superconducting properties are enhanced on either side of the Dirac node. The use of the theoretical analysis to make statements like "The dip in the $T_c \sim x$ phase diagram can be understood as a consequence of the complex competition between the RKKY interaction and antiferromagnetic exchange coupling" is hard to see. The authors need to be more clear about the theory, are the parameters used extracted from data? Are they physically relevant? It is hard to make conclusive claims when this is not known. Can the authors perform experiments or first principles calculations to provide physically relevant parameters such as J to the RKKY model? For example I can imagine a similar scenario resulting from spin split Rashba states at the interface of the TI and FeTe.
- Are the authors concerned about the influence of indium contacts regarding superconductivity?
- Do the authors have hall data where they can compare kF from Dirac cone in ARPES to transport
- Based on the EDS maps there seems to be Sb diffusion into FeTe for Sb_2Te_3 . Can the authors comment on if this? Could this lead to the decrease in T_c or H_c with increasing Sb concentration?
- Extended data fig 4 the atom labels should be more clear aka show also where Sb sits
- Arpes data, resolution ok by looking at deep states. Might good to see better resolution, can the authors measure cold? Especially since the transport results are for below the Neel temperature. At least could show second derivative of data
- ARPES Fermi map could show twinning, do the authors have this data?
- Can the authors perform magnetic measurements (susceptibility) to understand the magnetic ordering pre and post deposition of the TI layer? This will help strengthen "our transport results indicate that the long-range antiferromagnetic order is weakened or even destroyed after the deposition of the $(Bi_{1-x}Sb_x)_2Te_3$ layer"
- Comment on surface state sensitivity using He1alpha (21 eV) light
- Show picture of hall bar device measured
- What are the size of the twin domains and do the boundaries have an effect on transport? The authors should comment on strain induced in the TI layer by the lattice mismatch, does this influence the topological properties? XRD RSM data could be useful.
- Comment on why the lower half of the Dirac cone is less pronounced in the MDC's at higher x . The authors should mention if the lines are fits to the data or to guide the eye.

If the authors can provide additional details to the points I make above (or make clearer) in the paper, then I can recommend this paper for publication in Nature Communications.

Reviewer #2 (Remarks to the Author):

The superconductivity that emerges when FeTe is interfaced with (Bi,Sb)₂Te₃, both non-superconducting materials, is considered a promising platform for studying TSC in 2D. The new aspect in the present manuscript is tuning the chemical potential of the TI layer by changing the Bi/Sb ratio.

The authors use MBE to grow the heterostructure, in-situ ARPES to measure the chemical potential and ex-situ resistivity magnetoresistance measurements to characterize the SC state.

The MBE grown devices are of high quality as evidenced by the TEM data. The ARPES data, on the other hand, is not of very high quality although it allow a good measurement of the position of the Dirac point and of the chemical potential. The weakest part, in my opinion, are the transport measurements. All the data are normalized, the authors mention in the text that they have sheet resistance data for the samples but it is not shown in the manuscript. No Hall data are shown and the charge carrier density and its dependence on the Bi/Sb ratio is not shown.

The main result is a correlation found between the transition temperature and the chemical potential. It is claimed that since there is a minimum in T_c at the Bi/Sb level for which the Dirac-point is located at the Fermi level it must be that the surface-state electrons play a role in the interfacial SC.

I find this observation very circumstantial and I don't think it advances significantly our understanding of the physics of this interesting system. I can not recommend publication of this manuscript in Nature Comm.

Below I list a few questions/comments:

1) How do we know the SC resides only on the interface? The (Bi,Sb)₂Te₃ layer in these devices is metallic, so the system contains 4 conducting channels in parallel: The FeTe layer, the surface state on the interface, the TI bulk and the upper surface state. SC pairing can emerge in any of these, and in general all should develop at least some SC correlations due to proximity.

2) When the chemical potential is changed, the Dirac point moves but also the carrier density in the bulk changes. So the change in T_c could be related to the bulk properties and not to the Dirac surface states.

3) As the authors explain there is charge transfer between the FeSe, since the TI can provide significant screening can it be that the Dirac points on the surface and at the interface are not the same? In this case the minimum in T_c is found for a Bi/Sb level for which the Dirac point of the interface surface state is not at the Fermi level.

Reviewer #3 (Remarks to the Author):

In this work, the authors have prepared the FeTe/(Bi,Sb)₂Te₃ heterostructure, and found the interfacial superconductivity is strongly influenced by the ratio of Bi/Sb. They found when the Fermi level is tuned toward the surface Dirac point by controlling the Bi/Sb ratio, the superconductivity is suppressed. They attributed this phenomena to the RKKY interaction mediated by the Dirac fermions, which is large when the E_F away from Dirac point, and suppresses the antiferromagnetic order in FeTe and hence allows the superconductivity. Overall, I find this work interesting and will be helpful to realize topological superconductivity. However, there are some unclear points need to be clarified. I list my comment below.

1. In Fig. 1c, there is a metal-insulator transition associated with AFM-PM transition in pure FeTe. This transition is absent in FeTe/TI heterostructure. Is the FeTe still antiferromagnetic in this heterostructure? What is the T_N of it?

2. In a TI/magnet heterostructure, the surface Dirac cone is expected to be gapped due to the breaking of time reversal symmetry by the magnet. In this work, the authors performed the ARPES measurement at a temperature well above the T_N of FeTe. Could the authors perform ARPES measurement at low temperatures to see if there is a gap opening?

3. And if there is a gap opening of the Dirac cone, there should be some region where RKKY is invariable due to the vanishing Dirac Fermions. I will then expect some plateaus in Fig. 4(a-c) around $x=0.8$.

4. In principle the strength of RKKY can be quantitatively estimated. However, in Fig. 4e, there unit of J_{RKKY} is arb. Units. It's hard to evaluate whether the RKKY is still strong around T_c .

-----**Response to reviewers' comments**-----

Reviewer #1

H. Yi, et al grow TI/Fe chalcogenide epitaxial heterostructures to study interfacial superconductivity and the role of Dirac fermions. The central claim of the paper is that by changing the composition of the TI the Fermi level moves through the Dirac node which leads to a decrease in T_c on either side of the Dirac node suggesting Dirac fermions promote the interfacial superconductivity. This result leads to new platforms to study topological superconductivity, especially given the consensus around super-semi Majoranas is not agreed upon. The authors present a rather clear picture of the data, though some of it could be cleaned up and made more convincing. Below is a list of points that should be addressed.

We thank Reviewer #1 for his/her concise summary and positive assessment of our work.

Comment 1:

Main Points:

- The experimental evidence seems to support the statement that the superconducting properties are enhanced on either side of the Dirac node. The use of the theoretical analysis to make statements like “The dip in the T_c - x phase diagram can be understood as a consequence of the complex competition between the RKKY interaction and antiferromagnetic exchange coupling” is hard to see. The authors need to be more clear about the theory, are the parameters used extracted from data? Are they physically relevant? It is hard to make conclusive claims when this is not known. Can the authors perform experiments or first principles calculations to provide physically relevant parameters such as J to the RKKY model? For example I can imagine a similar scenario resulting from spin-split Rashba states at the interface of the TI and FeTe.

Response: We thank Reviewer #1 for the various suggestions in improving our discussion of the theory in the paper. Perhaps it will be useful to clarify the logic that links our experimental observations to the theoretical analysis. In accordance with the suggestions of Reviewer #1, we performed more theoretical calculations and provided more estimates to strengthen the connection between our theory and experimental findings.

As mentioned by Reviewer #1, the major experimental observation is that the superconducting T_c is enhanced on either side of the Dirac cone but suppressed when the chemical potential is near

the Dirac point. This observation implies that T_c is correlated with the Fermi momentum k_F of Dirac surface states in $(\text{Bi}_{1-x}\text{Sb}_x)_2\text{Te}_3$. As discussed in our manuscript, there are two main mechanisms for interface superconductivity: (i) the charge transfer effect between $(\text{Bi}_{1-x}\text{Sb}_x)_2\text{Te}_3$ and FeTe layers, and (ii) the suppression of long-range antiferromagnetic order in the FeTe layer. The charge transfer picture cannot explain our experimental observations because the transferred carriers should vary monotonically while tuning x in $(\text{Bi}_{1-x}\text{Sb}_x)_2\text{Te}_3$. On the other hand, strong spin-orbit coupling of Dirac surface states in TI can give rise to an RKKY-type magnetic interaction. This interaction can generally suppress antiferromagnetic order and is also potentially correlated with k_F . This motivates us to explore the k_F dependence of the RKKY interaction in the surface states in this work.

To investigate the effect of RKKY interaction on superconducting T_c in $(\text{Bi,Sb})_2\text{Te}_3/\text{FeTe}$ heterostructures, we first calculated the z -component RKKY strength J_z^{RKKY} of the Dirac surface states as a function of dimensionless variable $z = k_F R_{ij}$, where k_F is the Fermi momentum and R_{ij} is the distance between local magnetic moments [Liu et. al, *Phys. Rev. Lett.* **102**, 156603(2009)]. We found $J_z^{\text{RKKY}} < 0$ is the ferromagnetic type for a small z , and its magnitude almost reaches maximal when $k_F \rightarrow 0$. In our experiments, z varies in the range of $0 \sim 0.5$ as $R_{ij} \sim 5 \text{ \AA}$ is fixed, and k_F can be tuned by the Sb concentration x from $\sim 0.1 \text{ \AA}^{-1}$ to the Dirac point with $k_F \sim 0$. We noted that the parameters used in our theoretical calculations, including the Fermi velocity of Dirac cone A , k_F at different Sb concentration x , and the distance between the next nearest neighbor Fe atoms, are extracted from our experimental results. After obtaining J_z^{RKKY} , we took the standard spin-fluctuation theory for the unconventional superconductivity in iron-based superconductors and approximately derive the mean-field level T_c that is proportional to the total spin-spin interaction between the next nearest neighbor Fe atoms, i.e., $J = J_2 + J_z^{\text{RKKY}}$ where J_2 is the intrinsic/bare antiferromagnetic coupling. With the dependence of J_z^{RKKY} on k_F , the overall spin-spin interaction J is minimal at Dirac point $k_F \sim 0$ and increases for both electron- and hole-doped sides. As T_c is determined by the value of J , the non-monotonic behavior of J_z^{RKKY} can quantitatively explain the dip feature in the $T_c \sim x$ phase diagram in our experiments.

As noted above, all parameters used in our theoretical calculations are extracted from our experimental data. The arbitrary unit of J_z^{RKKY} is in the unit of a constant $\frac{J_z^2 V_0^2}{A R_{ij}^3}$, where J_z is the spin-

spin interaction strength between Dirac electron and local magnetic moment and V_0 is the in-plane area of the TI unit cell. We found that the quantitative results of J_z^{RKKY} depend only on k_F , as the integral part replies exclusively on the dimensionless parameter $z = k_F R_{ij}$ with fixed $R_{ij} = \sqrt{2} a_{FeTe}$ and $a_{FeTe} = 3.78 \text{ \AA}$. For different Sb concentrations x , the values of k_F are extracted from our ARPES band maps in Fig. 2 of the main text (Table 1). Therefore, we plotted J_z^{RKKY} in the unit of $\frac{J_z^2 V_0^2}{AR_{ij}^3}$ (Fig. 4 of the main text), which is directly associated with the non-monotonic feature of the superconducting T_c as a function of x .

Table 1 | The k_F values of 8QL $(\text{Bi}_{1-x}\text{Sb}_x)_2\text{Te}_3$ with different x from our ARPES band maps.

x	0	0.2	0.4	0.6	0.7	0.8	1
k_F (1/Å)	-0.106±0.01	-0.082±0.01	-0.054±0.02	-0.025±0.025	0.015±0.025	-0.005±0.025	0.034±0.025

Next, we followed Reviewer #1's suggestions and estimated the value of J_z^{RKKY} when $k_F \rightarrow 0$. Based on our ARPES band maps, we obtained the Fermi velocity $A \approx 3.5 \text{ eV} \cdot \text{\AA}$. The area $V_0 = \frac{3\sqrt{3}}{2} a_0^2 \approx 49.84 \text{ \AA}^2$ and the distance $R_{ij} \approx 5.34 \text{ \AA}$. However, it is difficult to obtain the spin-spin interaction strength J_z from the first-principals calculations for our $(\text{Bi,Sb})_2\text{Te}_3/\text{FeTe}$ heterostructures. For $J_z \in [10, 100] \text{ meV}$, the ferromagnetic-type RKKY strength J_z^{RKKY} is estimated to be $\sim - [0.02, 1.9] \text{ meV}$ at $k_F \rightarrow 0$. This value is in the order of 1 meV , which can greatly reduce the antiferromagnetic coupling in FeTe, whose energy scale can be estimated from the critical temperature of the bi-colinear AFM order (Néel temperature $T_N \sim 68 \text{ K}$) [Dai, *Rev. Mod. Phys.* **87**, 855(2015); Lipscombe et al, *Phys. Rev. Lett.* **106**, 057004 (2011)]. Therefore, it can significantly affect T_c of the interface-induced superconductivity in our $(\text{Bi,Sb})_2\text{Te}_3/\text{FeTe}$ heterostructures.

We agree with Reviewer #1 that it is an interesting topic to experimentally explore interfacial superconductivity in heterostructures formed by a Rashba material and FeTe. We noted that there are two Fermi surfaces in a Rashba model, but only one Fermi surface for one Dirac surface state of TI. This fact may lead to some differences between the Rashba model and Dirac surface state model, particularly when the chemical potential is near the Dirac point.

We added the relevant discussion in the revised manuscript.

Comment 2:

- Are the authors concerned about the influence of indium contacts regarding superconductivity?

Response: The superconducting temperature T_c of indium contacts is ~ 3.4 K, which is lower than the $T_{c,0}$ values of most samples used in our work (specifically, for $x < 0.8$ and $x > 0.85$). We also note that all experimental data show no hint of any anomaly at or near ~ 3.4 K. Moreover, the values of the sheet longitudinal resistance R for all samples used in this work were obtained through four-terminal measurements, rather than two-terminal measurements. These two aspects suggest that the emergent interface-induced superconductivity realized in $(\text{Bi}_{1-x}\text{Sb}_x)_2\text{Te}_3/\text{FeTe}$ heterostructures cannot be attributed to the presence of the indium contacts.

To exclude the potential influence of indium contacts on superconductivity in our $(\text{Bi}_{1-x}\text{Sb}_x)_2\text{Te}_3/\text{FeTe}$ heterostructures, we used silver epoxy (EPO-TEK H20E) and indium dots to make contacts on two Hall bar devices from the same 8QL $\text{Bi}_2\text{Te}_3/50$ UC FeTe heterostructure and performed transport measurements. We found that these two devices show similar R - T curves (Fig. R1). Therefore, the indium contacts used in this work do not contribute to the formation of the interface-induced superconductivity in our $(\text{Bi}_{1-x}\text{Sb}_x)_2\text{Te}_3/\text{FeTe}$ heterostructures. We added Fig. R1 in Supplementary Information.

Fig. R1| Temperature dependence of the sheet longitudinal resistance R of 8 QL $\text{Bi}_2\text{Te}_3/50$ UC FeTe measured with indium and silver epoxy contacts. The electrical contacts of the Hall bar devices are made of pressed indium dots (blue) and silver epoxy (red).

Comment 3:

- Do the authors have hall data where they can compare k_F from Dirac cone in ARPES to transport

Response: Yes, we did perform transport measurements on our $(\text{Bi}_{1-x}\text{Sb}_x)_2\text{Te}_3/\text{FeTe}$ heterostructures. However, as noted in our manuscript, FeTe is an antiferromagnetic metal, which exhibits higher conductivity compared to the $(\text{Bi}_{1-x}\text{Sb}_x)_2\text{Te}_3$ layer. In addition, the Hall trace for the FeTe layer displays a nearly zero slope at $T = 2$ K (Fig. R2), which indicates a significantly higher carrier density in the FeTe layer. Therefore, the Hall trace of the $(\text{Bi}_{1-x}\text{Sb}_x)_2\text{Te}_3/\text{FeTe}$ heterostructure does not accurately reflect the carrier density or the position of the chemical potential in the $(\text{Bi}_{1-x}\text{Sb}_x)_2\text{Te}_3$ layer. For the $(\text{Bi}_{1-x}\text{Sb}_x)_2\text{Te}_3$ layer without the FeTe layer, a comprehensive analysis comparing k_F derived from ARPES and Hall transport measurements has been carefully studied in our prior work [Zhang et. al, *Nature Commun.* **2**, 574 (2011)]. Therefore, for the $(\text{Bi}_{1-x}\text{Sb}_x)_2\text{Te}_3/\text{FeTe}$ heterostructures used in this work, the k_F value obtained from ARPES measurements is considered more reliable. We added Fig. R2 in Supplementary Information.

Fig. R2| Hall traces of 50 UC FeTe/SrTiO₃(100) at $T = 2$ K.

Comment 4:

- Based on the EDS maps there seems to be Sb diffusion into FeTe for Sb₂Te₃. Can the authors comment on if this? Could this lead to the decrease in T_c or H_c with increasing Sb concentration?

Response: We thank Reviewer #1 for bringing up this issue. We think the apparent trace Sb signal in the FeTe layer is from Te atoms rather than diffused Sb atoms. In EDS spectra of Sb₂Te₃ and FeTe layers (Fig. R3), Sb and Te show very similar peak positions. The Te $L_{\alpha 1}$ peak is located near the Sb $L_{\beta 1}$ peak, and the Te $L_{\beta 1}$ peaks overlap with the Sb $L_{\beta 2}$ peaks. These Te L peaks contribute

to the observed background Sb signal in the FeTe layer. Moreover, for $x > 0.85$, T_c increases with increasing x . This observation also suggests that the decreases in T_c and $\mu_0 H_c$ are not caused by the Sb diffusion into the FeTe layer.

Fig. R3| EDS spectra of the Sb₂Te₃ and FeTe layers in the 8 QL Sb₂Te₃/50 UC FeTe heterostructure. The Sb and Te peaks in the Sb₂Te₃ layer (top), and the Te peaks in the FeTe layer (bottom).

Comment 5:

- Extended data fig 4 the atom labels should be more clear aka show also where Sb sits

Response: Done.

Comment 6:

- Arpes data, resolution ok by looking at deep states. Might good to see better resolution, can the authors measure cold? Especially since the transport results are for below the Neel temperature. At least could show second derivative of data

Response: We apologize to Reviewer #1 for not including the low-temperature ARPES in our original manuscript. A similar comment is also made by Reviewer #3 (**Comment 2** of Reviewer #3). We followed the suggestions from Reviewers #1 and #3 and performed ARPES measurements on the 8 QL Bi₂Te₃/50 UC FeTe heterostructure at liquid nitrogen and liquid helium temperatures.

Fig. R4| Dirac surface states in the 8 QL $\text{Bi}_2\text{Te}_3/50$ UC FeTe heterostructures at different temperatures. **a-c**, ARPES band maps of 8 QL $\text{Bi}_2\text{Te}_3/50$ UC FeTe sample measured at room temperature ($T \sim 300$ K, **a**), liquid nitrogen temperature ($T \sim 77$ K, **b**), and liquid helium temperature ($T \sim 10$ K, **c**). The red and blue dashed lines indicate the Dirac surface states. **d**, Second derivative plot of ARPES band map in (**c**).

Fig. R5| Dirac surface states in 8 QL $(\text{Bi}_{1-x}\text{Sb}_x)_2\text{Te}_3/50$ UC FeTe heterostructures with different Sb concentrations x . **a-g**, Second derivative plots of the ARPES band map in Fig. 2 of the main text.

We found that the 8 QL $\text{Bi}_2\text{Te}_3/50$ UC FeTe heterostructure shows a similar band structure at low and room temperatures, in particular a similar Fermi momentum k_F (Fig. R4). We note that one prior ARPES measurement was conducted on $\text{Bi}_2\text{Te}_3/\text{FeTe}$ heterostructures at $T = 30$ K [Kenta Owada et. al, *Phys. Rev. B* **100**, 064518 (2019)]. The Dirac surface states of Bi_2Te_3 are gapless below the Néel temperature ($T_N \sim 68$ K) of FeTe [Dai, *Rev. Mod. Phys.* **87**, 855(2015); Lipscombe et al, *Phys. Rev. Lett.* **106**, 057004 (2011)]. Furthermore, our prior study [Zhang et. al, *Nature Commun.* **2**, 574 (2011)] has demonstrated that the k_F value derived from room-temperature

ARPES band spectra is consistent with that from the low-temperature Hall transport measurements. Therefore, it is reasonable to use the room temperature ARPES band spectra to estimate the k_F values of the top $(\text{Bi}_{1-x}\text{Sb}_x)_2\text{Te}_3$ layer in this work.

We followed Reviewer #1's suggestion and acquired the second derivative band spectra of the ARPES results shown in Fig. 2 of the main text. With increasing x , the chemical potential is tuned from above to below the Dirac point (Fig. R5), which is consistent with our analysis in the original manuscript (Fig. 2). We added Figs. R4 and R5 in Supplementary Information.

Comment 7:

- ARPES Fermi map could show twinning, do the authors have this data?

Response: Yes. Figure R6 shows the ARPES Fermi surface map of the 8 QL $\text{Bi}_2\text{Te}_3/50$ UC FeTe heterostructure. Unlike the Fermi surface of Bi_2Te_3 single crystal [Chen et al, *Science* 325, 178 (2009)], which has a six-fold symmetry, the Bi_2Te_3 layer grown on the FeTe layer shows a twelve-fold symmetry (Fig. R6b). This observation indicates the existence of twin domains in our $\text{Bi}_2\text{Te}_3/\text{FeTe}$ heterostructures [Owada et. al, *Phys. Rev. B* 100, 064518 (2019)]. We added Fig. R6 in Supplementary Information.

Fig. R6| Constant energy contours of the 8 QL $\text{Bi}_2\text{Te}_3/50$ UC FeTe heterostructure. a, $E-E_F=0$ eV (i.e. Fermi surface). b, $E-E_F=-0.23$ eV. The ARPES spectra in (b) are composed of both Dirac surface states and bulk valence bands. The appearance of twelve-fold symmetry in the bulk states indicates the existence of the twin domains in the 8 QL Bi_2Te_3 layer.

Comment 8:

- Can the authors perform magnetic measurements (susceptibility) to understand the magnetic ordering pre and post deposition of the TI layer? This will help strengthen “our transport results

indicate that the long-range antiferromagnetic order is weakened or even destroyed after the deposition of the $(\text{Bi}_{1-x}\text{Sb}_x)_2\text{Te}_3$ layer”

Response: We thank Reviewer #1 for raising this question. We did perform superconducting quantum interference device (SQUID) measurements on FeTe layers with and without the TI layer. However, because of the small volume of FeTe and its bicollinear AFM nature, we could not determine the T_N value of the FeTe layers with and without the TI layer through SQUID measurements. Moreover, the suppression of the bicollinear AFM order in FeTe is primarily localized near the interface. Therefore, this property makes SQUID measurements impractical for examining the interface-induced suppression of the bicollinear AFM order in FeTe. We are performing and analyzing polarized neutron reflectometry (PNR) measurements on $(\text{Bi}_{1-x}\text{Sb}_x)_2\text{Te}_3/\text{FeTe}$ heterostructures and exploring the T_N change after the deposition of the $(\text{Bi}_{1-x}\text{Sb}_x)_2\text{Te}_3$ layer. The PNR results will be reported in a separate paper soon.

Comment 9:

- Comment on surface state sensitivity using HeIalpha (21 eV) light

Response: Based on the well-established empirical curve about the inelastic mean free paths of electrons as a function of energy above the Fermi level, He I_α light with an energy of ~21 eV is more surface sensitive [Seah and Dench, *Surf. Interface Anal.* **1**, 2-11(1979)]. We added this information in the Method section of the revised manuscript.

Fig. R7| Photograph of the $(\text{Bi,Sb})_2\text{Te}_3/\text{FeTe}$ Hall bar device used in our transport measurements.

Comment 10:

- Show picture of hall bar device measured

Response: We added a photograph of the $(\text{Bi,Sb})_2\text{Te}_3/\text{FeTe}$ Hall bar device (Fig. R7) in Supplementary Information.

Comment 11:

- What are the size of the twin domains and do the boundaries have an effect on transport? The authors should comment on strain induced in the TI layer by the lattice mismatch, does this influence the topological properties? XRD RSM data could be useful.

Fig. R8| Atomic force microscopy image of the 8 QL $\text{Bi}_2\text{Te}_3/50$ UC FeTe heterostructure. The triangular pyramidal structures indicate two different domain orientations.

Response: To estimate the size of twin domains, we performed atomic force microscopy measurements on an 8 QL $\text{Bi}_2\text{Te}_3/50$ UC FeTe heterostructure (Fig. R8). Based on our prior atomic force microscopy studies on MBE-grown Bi_2Te_3 films [Richardella et al, *APL Mater.* **3**, 083303 (2015)], the presence of triangular pyramidal structures with a 180° rotation indicates the existence of twin domains in the 8 QL $\text{Bi}_2\text{Te}_3/50$ UC FeTe heterostructure. The domain size of the 8 QL $\text{Bi}_2\text{Te}_3/50$ UC FeTe heterostructure is found to be ~ 500 nm, comparable to that of the MBE-grown Bi_2Te_3 films on sapphire (0001) [Richardella et al, *APL Mater.* **3**, 083303 (2015); Kriegner et al, *J. Appl. Crystallogr.* **50**, 369 (2017)]. The existence of domain boundaries in our TI/FeTe heterostructures does not affect the transport results because the interfacial superconductivity originates from the bottom FeTe layer.

Fig. R9 | XRD reciprocal space mapping (RSM) of the $\text{Bi}_2\text{Te}_3/\text{FeTe}$ and $\text{Sb}_2\text{Te}_3/\text{FeTe}$ heterostructures. **a**, Symmetric RSM around the Bi_2Te_3 (0 0 15) peak showing SrTiO_3 (0 0 2) and FeTe (0 0 3) reflections. The dotted lines show the extent of the scanned region in reciprocal space. **b**, Symmetric RSM measured at $\chi=58.08^\circ$ showing Bi_2Te_3 (0 1 5) peak partially overlapped with FeTe (1 0 1). **c**, ϕ scans of Bi_2Te_3 and SrTiO_3 . Bi_2Te_3 shows twin domains with a 30° rotation. **d**, RSM of Sb_2Te_3 (0 0 15) showing SrTiO_3 and FeTe as in (a). **e**, RSM measured at $\chi=58.78^\circ$ showing Sb_2Te_3 (0 1 5) at a larger Q_z than the corresponding Bi_2Te_3 peak due to the smaller in-plane lattice constant of Sb_2Te_3 . **f**, ϕ scans of Sb_2Te_3 , FeTe and SrTiO_3 . FeTe is epitaxial to SrTiO_3 substrate while Sb_2Te_3 shows twin domains with a 30° rotation.

We followed Reviewer #1's suggestions and performed XRD reciprocal space mapping (RSM) measurements on $\text{Bi}_2\text{Te}_3/\text{FeTe}$ and $\text{Sb}_2\text{Te}_3/\text{FeTe}$ heterostructures, respectively (Fig. R9). These two heterostructures were aligned using the $\text{SrTiO}_3(002)$ reflection and the azimuthal angle for off-axis peaks was aligned using the ϕ scans. The in-plane lattice constant was determined by measuring the off-axis peaks in the symmetric geometry after tilting the sample in χ because the reflections available for asymmetric scans had very low intensities. The 2θ positions for reflections were measured by fitting a Lorentzian function to the data. For the 8 QL $\text{Bi}_2\text{Te}_3/50$ UC FeTe heterostructure, the out-of-plane $\text{Bi}_2\text{Te}_3(0\ 0\ 15)$ and off-axis (0 1 5) reflections were measured

(Figs. R9a and R9b), which were used to determine the in-plane and out-of-plane lattice constants. The calculated strain in the Bi_2Te_3 layer is $\sim 1\%$ in-plane tensile strain, with no significant strain observed out of the plane. For the 8 QL $\text{Bi}_2\text{Te}_3/50$ UC FeTe heterostructure, a light strain is observed in the in-plane or out-of-plane directions ($\sim 0.3\%$) (Figs. R9d and R9e). The presence of small strains in the $(\text{Bi}_{1-x}\text{Sb}_x)_2\text{Te}_3$ layer grown on the FeTe layer implies no changes in the topological properties of $(\text{Bi}_{1-x}\text{Sb}_x)_2\text{Te}_3$ [Liu et al, *Acta Crystallogr. C Struct. Chem.* **70**, 118 (2014); Aramberri et al, *Phys. Rev. B* **95**, 205422 (2017)]. Moreover, the twin domains with a 30° rotation are further verified by ϕ scans of $(\text{Bi}_{1-x}\text{Sb}_x)_2\text{Te}_3/50$ UC FeTe heterostructures (Figs. R9c and R9f), which are consistent with our RHEED, atomic force microscopy, and ARPES measurements (Supplementary Figs. 1, 5, and 6).

Comment 12:

- Comment on why the lower half of the Dirac cone is less pronounced in the MDC's at higher x. The authors should mention if the lines are fits to the data or to guide the eye.

Response: We thank Reviewer #1 for bringing up this issue. In the literature, it has been consistently observed that within the Bi_2Te_3 family TI (i.e., Bi_2Te_3 , Bi_2Se_3 , Sb_2Te_3 , and their compounds), the lower half of the Dirac cone is usually less pronounced than its top half [Xia et al, *Nature Phys.* **5**, 398-402 (2009); Chen et al, *Science* **325**, 178-181 (2009); Hsieh et al, *Phys. Rev. Lett.* **103**, 146401 (2009); Zhang et al, *Nature Phys.* **6**, 584 (2010)]. The underlying mechanism for this phenomenon is still not clear. In our experiments, with increasing x , the bulk valence bands appear. This may lower the intensity contrast between the bulk bands and the lower half of the Dirac surface states at higher x .

The dashed lines in the top panel of Fig. 2 are used to guide the eye. The dashed lines in the bottom panel of Fig. 2 show the peak locations in each momentum distribution curve. We added this information in the caption of Fig. 2.

If the authors can provide additional details to the points I make above (or make clearer) in the paper, then I can recommend this paper for publication in Nature Communications.

We followed the suggestions from the three reviewers and carefully revised our manuscript. We hope the revised manuscript meets Reviewer #1's standard/criterion for publication in *Nature Communications*.

Reviewer #2

The superconductivity that emerges when FeTe is interfaced with (Bi,Sb)₂Te₃, both non-superconducting materials, is considered a promising platform for studying TSC in 2D.

The new aspect in the present manuscript is tuning the chemical potential of the TI layer by changing the Bi/Sb ratio.

The authors use MBE to grow the heterostructure, in-situ ARPES to measure the chemical potential and ex-situ resistivity magnetoresistance measurements to characterize the SC state.

We thank Reviewer #2 for concisely summarizing our work and highlighting the importance of our work. We revised our manuscript according to the suggestions from the three reviewers and hope the revised manuscript meets his/her standard for *Nature Communications*.

Comment 1:

The MBE grown devices are of high quality as evidenced by the TEM data. The ARPES data, on the other hand, is not of very high quality although it allow a good measurement of the position of the Dirac point and of the chemical potential. The weakest part, in my opinion, are the transport measurements. All the data are normalized, the authors mention in the text that they have sheet resistance data for the samples but it is not shown in the manuscript. No Hall data are shown and the charge carrier density and its dependence on the Bi/Sb ratio is not shown.

Response: Reviewer #2's comments on MBE growth and ARPES data are fair and reasonable. However, we are puzzled by his/her comment "*All the data are normalized, the authors mention in the text that they have sheet resistance data for the samples but it is not shown in the manuscript*". We noted that all *R-T* curves shown in our manuscript (Figs. 1c and 3 of the main text) are original and not normalized.

As noted in our response to **Comment 3** of Reviewer #1, FeTe is an antiferromagnetic metal, which exhibits higher conductivity compared to the (Bi_{1-x}Sb_x)₂Te₃ layer. Therefore, the Hall trace of the (Bi_{1-x}Sb_x)₂Te₃/FeTe heterostructure does not accurately reflect the carrier density (or the position of the chemical potential) in the (Bi_{1-x}Sb_x)₂Te₃ layer. For the (Bi_{1-x}Sb_x)₂Te₃ layer without the FeTe layer, a comprehensive analysis comparing k_F derived from ARPES and Hall transport measurements has been carefully studied in our prior work [Zhang et. al, *Nature Commun.* **2**, 574 (2011)]. Therefore, for the (Bi_{1-x}Sb_x)₂Te₃/FeTe heterostructures used in this work, we cannot

use the carrier density obtained from Hall measurements to estimate the value of k_F . See more discussion in our response to **Comment 3** of Reviewer #1.

Comment 2:

The main result is a correlation found between the transition temperature and the chemical potential. It is claimed that since there is a minimum in T_c at the Bi/Sb level for which the Dirac point is located at the Fermi level it must be that the surface-state electrons play a role in the interfacial SC.

I find this observation very circumstantial and I don't think it advances significantly our understanding of the physics of this interesting system. I can not recommend publication of this manuscript in Nature Comm.

Response: We thank Reviewer #2 for his/her summary of our manuscript. We respectfully disagree with Reviewer #2's comment "*I find this observation very circumstantial and I don't think it advances significantly our understanding of the physics of this interesting system.*". Reviewer #2 concluded that our observation is "*very circumstantial*". However, Reviewer #2 did not provide us with any specific details or explanation to support this conclusion. As noted in our manuscript, our experiments demonstrate that the Dirac electrons of the TI layer participate in the formation of the interfacial superconductivity in $(\text{Bi}_{1-x}\text{Sb}_x)_2\text{Te}_3/\text{FeTe}$ heterostructures. This observation provides strong motivation for employing the $(\text{Bi}_{1-x}\text{Sb}_x)_2\text{Te}_3/\text{FeTe}$ heterostructure as a model material system to explore Majorana physics and topological quantum computation. We noted that this novel aspect has been recognized by both Reviewers #1 and 3.

(i) From Reviewer #1 "*This result leads to new platforms to study topological superconductivity, especially given the consensus around super-semi Majoranas is not agreed upon.*".

(ii) From Reviewer #3 "*Overall, I find this work interesting and will be helpful to realize topological superconductivity.*".

We followed the suggestions from the three reviewers and have carefully revised our manuscript. We hope the revised manuscript meets Reviewer #2's standard/criterion for *Nature Communications*.

Comment 3:

Below I list a few questions/comments:

1) How do we know the SC resides only on the interface? The $(\text{Bi,Sb})_2\text{Te}_3$ layer in these devices is metallic, so the system contains 4 conducting channels in parallel: The FeTe layer, the surface state on the interface, the TI bulk and the upper surface state. SC pairing can emerge in any of these, and in general all should develop at least some SC correlations due to proximity.

Response: We are puzzled by Reviewer #2's question "*How do we know the SC resides only on the interface?*". In our manuscript, we never made this claim. Before we directly address this comment, we first would like to clarify one common misunderstanding on interfacial superconductivity. The interfacial superconductor does not mean that the superconductivity exists between two mono-atomic layers adjacent to the interface. So far, most known interfacial superconductors involve charge transfer between a parent compound and a suitable partner material. The parent compound itself does not need to be a superconductor *per se*, but can become one by charge transfer and/or strain effect, such as SrTiO_3 in $\text{SrTiO}_3/\text{LaAlO}_3$ [Reyren et al, *Science* **317**, 1196 (2007)] and La_2CuO_4 in $\text{La}_{1.55}\text{Sr}_{0.45}\text{CuO}_4/\text{La}_2\text{CuO}_4$ [Gozar et al, *Nature* **455**, 782(2008)]. In all these cases, the formation of the interfacial superconductivity needs multiple layers away from the interface.

In our $(\text{Bi}_{1-x}\text{Sb}_x)_2\text{Te}_3/\text{FeTe}$ heterostructures, the parent compound is FeTe. The FeTe layer can become a superconductor when its antiferromagnetic order is suppressed [Mizuguchi et al, *Appl. Phys. Lett.* **94**, 012503 (2009); Han et al, *Phys. Rev. Lett.* **104**, 017003 (2010); Dai, *Rev. Mod. Phys.* **87**, 855(2015)]. Prior studies have unambiguously demonstrated that superconductivity emerges only when the Bi_2Te_3 layer is deposited on top of the FeTe layer [He et. al, *Nature Commun.* **5**, 4247 (2014); Yasuda et. al, *Nature Commun.* **10**, 2734 (2019); Qin et al, *Nano Lett.* **20**, 3160 (2020)]. In our work, we attributed the suppression of the antiferromagnetic order in the FeTe layer to the Dirac surface states mediated RKKY interaction, which leads to the FeTe layer acquiring superconductivity. Moreover, as noted in our manuscript, the proximity effect-induced superconducting gap has been observed on the top surface of the $\text{Bi}_2\text{Te}_3/\text{FeTe}$ heterostructures through scanning tunneling microscopy and spectroscopy (STM/S) measurements [Qin et al, *Nano Lett.* **20**, 3160 (2020)]. It appears that Reviewer #2 might have misunderstood the location of the interface-induced superconductivity in our $(\text{Bi}_{1-x}\text{Sb}_x)_2\text{Te}_3/\text{FeTe}$ heterostructures.

Comment 4:

2) When the chemical potential is changed, the Dirac point moves but also the carrier density in

the bulk changes. So the change in T_c could be related to the bulk properties and not to the Dirac surface states.

Response: We are puzzled by Reviewer #2's comment "*When the chemical potential is changed, the Dirac point moves but also the carrier density in the bulk changes.*". The band structure of $(\text{Bi}_{1-x}\text{Sb}_x)_2\text{Te}_3$ has been well studied in prior ARPES studies [Zhang et al, *Nature Commun.* **2**, 574 (2011); Kong et al, *Nature Nanotechnol.* **6**, 705(2011)]. The bulk band gap of $(\text{Bi}_{1-x}\text{Sb}_x)_2\text{Te}_3$ is found to be $\sim 300\text{meV}$. **Figure R10** shows the schematic band structure of $(\text{Bi}_{1-x}\text{Sb}_x)_2\text{Te}_3$ based on our ARPES results. As noted in our manuscript, for the $x = 0$ sample (i.e. Bi_2Te_3), the Dirac point is located at $\sim 245\text{meV}$ below the chemical potential. With increasing x , the chemical potential moves downward and gradually approaches the Dirac point. For the $x = 0.8$ sample, the chemical potential almost crosses the Dirac point. With further increasing x , the chemical potential is below the Dirac point for the $x = 1$ sample (**Fig. R10**). We noted that when x is very close to 1, the chemical potential may cross the bulk valence bands along the Γ -M direction [Zhang et al, *Nature Commun.* **2**, 574 (2011); Kong et al, *Nature Nanotechnol.* **6**, 705(2011)]. Therefore, for $x < 1$, the chemical potential change does not affect the bulk carrier density because the chemical potential does not cross the bulk bands. This is the primary reason we established a connection between Dirac electrons and interfacial superconductivity (**Comment 2** of Reviewer #2 above) and made the claim of Dirac-fermion-assisted interfacial superconductivity in our $(\text{Bi}_{1-x}\text{Sb}_x)_2\text{Te}_3/\text{FeTe}$ heterostructures.

Fig. R10| Schematic band structures of $(\text{Bi}_{1-x}\text{Sb}_x)_2\text{Te}_3$ based on our ARPES. The dashed line indicates the chemical potential of $(\text{Bi}_{1-x}\text{Sb}_x)_2\text{Te}_3$ with different x .

Comment 5:

3) As the authors explain there is charge transfer between the FeSe, since the TI can provide significant screening can it be that the Dirac points on the surface and at the interface are not the same? In this case the minimum in T_c is found for a Bi/Sb level for which the Dirac point of the interface surface state is not at the Fermi level.

Response: We thank Reviewer #2 for bringing up this issue. As noted in our manuscript, both work functions in Bi_2Te_3 (~5.3 eV) and Sb_2Te_3 (~5.0 eV) are greater than that of FeTe (4.4~4.8 eV) [Takane et al, *Appl. Phys. Lett.* **109**, 091601 (2016); Qin et al, *Nano Lett.* **20**, 3160 (2020); Owada et al, *Phys. Rev. B* **100**, 064518 (2019)]. This work function mismatch between $(\text{Bi}_{1-x}\text{Sb}_x)_2\text{Te}_3$ and FeTe is expected to cause a charge transfer. Prior studies [Qin et al, *Nano Lett.* **20**, 3160-3168 (2020); Owada et al, *Phys. Rev. B* **100**, 064518 (2019)] have claimed the occurrence of hole carrier transfer from Bi_2Te_3 to FeTe, which may give rise to interfacial superconductivity in the FeTe layer by screening out the strong Coulomb repulsion. In this work, besides charge transfer, the Dirac electrons-mediated RKKY interaction can also suppress the bi-collinear antiferromagnetic order, which leads to the FeTe layer acquiring superconductivity.

Reviewer #2 is correct that the charge transfer between $(\text{Bi}_{1-x}\text{Sb}_x)_2\text{Te}_3$ and FeTe can lead to the Dirac points on the top and bottom [i.e. the $(\text{Bi}_{1-x}\text{Sb}_x)_2\text{Te}_3/\text{FeTe}$ interface] surfaces of $(\text{Bi}_{1-x}\text{Sb}_x)_2\text{Te}_3$ at different energies. Because of this energy difference, the minimum value of T_c should be observed away from the Bi/Sb ratio at which the chemical potential crosses the Dirac point on the top surface of $(\text{Bi}_{1-x}\text{Sb}_x)_2\text{Te}_3$. In our experiments, the chemical potential crosses the Dirac point at $x=0.8$ (Fig. 4a of the main text). However, the minimum values of both $T_{c, \text{onset}}$ and $T_{c,0}$ were observed at $x=0.85$ (Fig. 4b of the main text). This observation further confirms that the Dirac points on the top and bottom surfaces of $(\text{Bi}_{1-x}\text{Sb}_x)_2\text{Te}_3$ are located at different energies. We added the relevant discussion in the revised manuscript.

Reviewer #3

In this work, the authors have prepared the $\text{FeTe}/(\text{Bi,Sb})_2\text{Te}_3$ heterostructure, and found the interfacial superconductivity is strongly influenced by the ratio of Bi/Sb. They found when the Fermi level is tuned toward the surface Dirac point by controlling the Bi/Sb ratio, the superconductivity is suppressed. They attributed this phenomena to the RKKY interaction

mediated by the Dirac fermions, which is large when the EF away from Dirac point, and suppresses the antiferromagnetic order in FeTe and hence allows the superconductivity. Overall, I find this work interesting and will be helpful to realize topological superconductivity. However, there are some unclear points need to be clarified. I list my comment below.

We thank Reviewer #3 for his/her positive assessment of our work and thoughtful comments below.

Comment 1:

1. In Fig. 1c, there is a metal-insulator transition associated with AFM-PM transition in pure FeTe. This transition is absent in FeTe/TI heterostructure. Is the FeTe still antiferromagnetic in this heterostructure? What is the T_N of it?

Fig. R11| Temperature dependence of the sheet longitudinal resistance R of 8 QL $(\text{Bi}_{1-x}\text{Sb}_x)_2\text{Te}_3/50$ UC FeTe heterostructures. a-h, $x=0$ (a), $x=0.2$ (b), $x=0.4$ (c), $x=0.7$ (d), $x=0.8$ (e), $x=0.85$ (f), $x=0.95$ (g), and $x=1$ (h). The arrows indicate the hump features in R - T curves of these 8 QL $(\text{Bi}_{1-x}\text{Sb}_x)_2\text{Te}_3/50$ UC FeTe heterostructures.

Response: We thank Reviewer #3 for raising this question. Upon the deposition of the TI layer onto the FeTe layer, the superconductivity appears and the antiferromagnetic hump feature becomes much broader (Fig. 1c of the main text and Fig. R11). Therefore, it is likely that the bicollinear antiferromagnetic order of the FeTe layer near the $(\text{Bi}_{1-x}\text{Sb}_x)_2\text{Te}_3/\text{FeTe}$ interface is suppressed [Dai, *Rev. Mod. Phys.* **87**, 855(2015)]. However, the FeTe layer near the SrTiO_3 (100) substrate may maintain its bicollinear antiferromagnetic order. Due to the much broader hump features, it is challenging to determine the T_N values of our $(\text{Bi}_{1-x}\text{Sb}_x)_2\text{Te}_3/\text{FeTe}$ heterostructures through their R - T curves. As noted in our response to **Comment 8** of Reviewer #1, now we are performing and analyzing PNR measurements on $(\text{Bi}_{1-x}\text{Sb}_x)_2\text{Te}_3/\text{FeTe}$ heterostructures and exploring the T_N change after the deposition of the $(\text{Bi}_{1-x}\text{Sb}_x)_2\text{Te}_3$ layer. The PNR results will be reported in a separate paper soon. We added Fig. R11 in Supplementary Information.

Comment 2:

2. In a TI/magnet heterostructure, the surface Dirac cone is expected to be gapped due to the breaking of time reversal symmetry by the magnet. In this work, the authors performed the ARPES measurement at a temperature well above the T_N of FeTe. Could the authors perform ARPES measurement at low temperatures to see if there is a gap opening?

Response: A similar comment was also made by Reviewer #1 (**Comment 6** of Reviewer #1). We followed the suggestion from Reviewers #1 and #3 and performed ARPES measurements on the 8 QL $\text{Bi}_2\text{Te}_3/50$ UC FeTe heterostructure at liquid nitrogen and liquid helium temperatures. We found the 8 QL $\text{Bi}_2\text{Te}_3/50$ UC FeTe heterostructure shows a similar band structure at low and room temperatures and there is no gap opening at the Dirac point (Fig. R4). Moreover, the prior ARPES measurement was conducted on $\text{Bi}_2\text{Te}_3/\text{FeTe}$ heterostructure at $T=30\text{K}$ [Kenta Owada et. al, *Phys. Rev. B* **100**, 064518 (2019)]. The Dirac surface states of Bi_2Te_3 are gapless below the T_N of FeTe.

As noted in our manuscript and our response to **Comment 1** of Reviewer #3, the appearance of the interface-induced superconductivity in our $(\text{Bi}_{1-x}\text{Sb}_x)_2\text{Te}_3/\text{FeTe}$ heterostructures is attributed to the suppression of the bicollinear antiferromagnetic order in the FeTe layer near the $(\text{Bi}_{1-x}\text{Sb}_x)_2\text{Te}_3/\text{FeTe}$ interface. In other words, when the TI layer is deposited onto the FeTe layer, the FeTe layer near the TI/FeTe interface will lose or weaken its antiferromagnetic order. It should be noted that that does not mean that the RKKY interaction can lead to ferromagnetism at the interface, because the induced RKKY interaction is weaker than the energy scale of anti-ferromagnetic order

in the FeTe layer, see our response to **Comment 1** of Reviewer #1. Even if the bicollinear antiferromagnetic order is maintained throughout the entire FeTe layer, the in-plane antiferromagnetic order cannot induce a gap opening at the Dirac point [Chang et al, *Science* **340**, 167 (2013); Chang, Liu, and MacDonald, *Rev. Mod. Phys.* **95**, 011002 (2023)]. Therefore, the absence of the gap opening in our low-temperature ARPES measurements is scientifically reasonable. See more discussion in our response to **Comment 6** of Reviewer #1.

Comment 3:

3. And if there is a gap opening of the Dirac cone, there should be some region where RKKY is invariable due to the vanishing Dirac Fermions. I will then expect some plateaus in Fig. 4(a-c) around $x=0.8$.

Response: As noted in our response to **Comment 2** of Reviewer #3 above, no gap opening is observed at the Dirac point in our $(\text{Bi}_{1-x}\text{Sb}_x)_2\text{Te}_3/\text{FeTe}$ heterostructures. In our response to **Comment 2** of Reviewer #3, we explained why the gap opening is absent in our $(\text{Bi}_{1-x}\text{Sb}_x)_2\text{Te}_3/\text{FeTe}$ heterostructures. We tend to agree with Reviewer #3 that if there is a gap opening in the TI layer, it is very likely the T_c values and the upper critical magnetic field $\mu_0 H_{c2,\perp}$ values would remain constant near the charge neutral point.

Comment 4:

4. In principle the strength of RKKY can be quantitatively estimated. However, in Fig. 4e, there unit of J_{RKKY} is arb. Units. It's hard to evaluate whether the RKKY is still strong around T_c .

Response: See our response to **Comment 1** of Reviewer #1.

-----List of changes-----

(All the changes in the main article are shown in blue)

1. Line 101 on Page 5, we rewrote the below sentence.

“Our MBE-grown $(\text{Bi}_{1-x}\text{Sb}_x)_2\text{Te}_3/\text{FeTe}$ films are characterized by scanning transmission electron microscopy (STEM) (Fig. 1 and Supplementary Figs. 2 to 4), atomic force microscopy (Supplementary Fig. 5), and ARPES (Fig. 2 and Supplementary Figs. 6 to 10) measurements.”

2. Line 148 on Page 7, we rewrote the below sentence.

“In all $(\text{Bi}_{1-x}\text{Sb}_x)_2\text{Te}_3/\text{FeTe}$ heterostructures with $0 \leq x \leq 1$, we observe linearly dispersed Dirac surface states at room temperature (Fig. 2 and Supplementary Fig. 8).”

3. Line 153 on Page 7, we added the below sentence.

“A similar Fermi momentum k_F is derived from ARPES band maps at low temperatures (Supplementary Fig. 10).”

4. Line 202 on Page 9, we rewrote the below sentences.

“This work function mismatch between $(\text{Bi}_{1-x}\text{Sb}_x)_2\text{Te}_3$ and FeTe is expected to cause a charge transfer, which may lead to the different energy levels for the Dirac points on the top and bottom [i.e. the $(\text{Bi}_{1-x}\text{Sb}_x)_2\text{Te}_3/\text{FeTe}$ interface] surfaces of $(\text{Bi}_{1-x}\text{Sb}_x)_2\text{Te}_3$. As a consequence, the minimum value of T_c is anticipated to be observed away from the Bi/Sb ratio at which the chemical potential crosses the Dirac point on the top surface of $(\text{Bi}_{1-x}\text{Sb}_x)_2\text{Te}_3$. In our experiments, the chemical potential crosses the Dirac point at $x=0.8$ (Fig. 4a). However, the minimum values of both $T_{c, \text{onset}}$ and $T_{c,0}$ are observed at $x=0.85$ (Fig. 4b). This observation further implies that the presence of band bending in the $(\text{Bi}_{1-x}\text{Sb}_x)_2\text{Te}_3$ layer.”

5. Line 252 on Page 12, we added the below sentence.

“We note that the parameters used in our theoretical calculations are extracted from our experiments. $|J_{\text{RKKY}}|$ is estimated to be ~ 1 meV for $k_F \rightarrow 0$, which is comparable to the intrinsic $J_2 \sim 10$ meV (Methods).”

6. Line 312 on Page 14, we added the below sentence.

“We note that He I_α light with an energy of ~ 21 eV is sensitive to the sample surface⁴⁹.”

7. Line 317 on Page 15, we rewrote the below sentence.

“We made the electrical contacts by pressing tiny indium spheres on the Hall bar, and the presence of the indium contacts does not influence the interface-induced superconducting in $(\text{Bi}_{1-x}\text{Sb}_x)_2\text{Te}_3/\text{FeTe}$ heterostructures (Supplementary Fig. 14).”

8. Line 323 on Page 15, we added the below paragraph.

“The Hall trace of the FeTe layer shows a nearly zero slope at $T = 2$ K (Supplementary Fig. 15), which indicates a significantly higher carrier density in the FeTe layer. Consequently, the Hall trace of the $(\text{Bi}_{1-x}\text{Sb}_x)_2\text{Te}_3/\text{FeTe}$ heterostructure does not accurately reflect the carrier density (or the position of the chemical potential) in the $(\text{Bi}_{1-x}\text{Sb}_x)_2\text{Te}_3$ layer. For the $(\text{Bi}_{1-x}\text{Sb}_x)_2\text{Te}_3$ layer without the FeTe layer, a comprehensive analysis comparing k_F derived from ARPES and Hall transport measurements has been carefully studied in our prior work³⁰. Therefore, for the $(\text{Bi}_{1-x}\text{Sb}_x)_2\text{Te}_3/\text{FeTe}$ heterostructures used in this work, the k_F value obtained from ARPES measurements is considered more reliable in reflecting the position of the chemical potential position of $(\text{Bi}_{1-x}\text{Sb}_x)_2\text{Te}_3$.”

9. Line 331 on Page 15, we added a section “**X-ray diffraction reciprocal space mapping (RSM) measurements**” in Method.

10. Line 373 on Page 17, we rewrote the below sentences.

“The RKKY interaction between \vec{S}_1 and \vec{S}_2 is given by $J_{RKKY} = \frac{J_{eff}^2 V_0^2}{AR_{ij}^3} \times \Phi_{zz}(k_F R_{ij})$, where V_0 is the in-plane area of the TI unit cell and R_{ij} is the in-plane spatial distance between the two magnetic moments³². Note that the constant coefficient $\frac{J_{eff}^2 V_0^2}{AR_{ij}^3}$ provides us the energy unit. The dimensionless integral part $\Phi_{zz}(z)$, as a function of dimensionless parameter $z = k_F R_{ij}$, is the addition of intra-band and inter-band contributions.”

11. Line 400 on Page 18, we rewrote the below sentences.

“Below we list the parameters used in our theoretical calculations. For different Sb concentrations x , the values of k_F are extracted from our ARPES band maps in Fig. 2 and are plotted in Fig. 4a. Based on our ARPES band maps, we obtained the Fermi velocity $A \approx 3.5 \text{ eV} \cdot \text{\AA}$. The area $V_0 = \frac{3\sqrt{3}}{2} a_0^2 \approx 49.84 \text{ \AA}^2$ and the distance $R_{ij} \approx 5.34 \text{ \AA}$. For $J_z \in [10, 100] \text{ meV}$, the ferromagnetic-type RKKY strength J_z^{RKKY} is estimated to be $\sim - [0.02, 1.9] \text{ meV}$ at $k_F \rightarrow 0$. This value is in the

order of 1 *meV*, which can greatly reduce the antiferromagnetic coupling in FeTe, whose energy scale can be estimated from the critical temperature of the bi-colinear AFM order ($T_N \sim 68$ K)^{52,53}. Therefore, it can significantly affect T_c of the interface-induced superconductivity in our (Bi,Sb)₂Te₃/FeTe heterostructures.”

12. Line 505 on Page 24, we added the below two sentences to the caption of Fig. 2.

“The dashed lines are used to guide the eye.”

“The dashed lines indicate the positions of the peaks in each momentum distribution curve.”

13. We moved all Extended Data Figures to Supplementary Information.

14. We added Supplementary Figs. 5, 6, 8, 10, 11, 13, 14, 15, and 16 in Supplementary Information.

15. We added seven references shown in blue in the revised manuscript.

16. We made numbers of minor revisions shown in blue in the revised manuscript and Supplementary Information.

REVIEWER COMMENTS

Reviewer #1 (Remarks to the Author):

The authors do a thorough job responding to my comments. I have a few follow up points, if these can be cleared up, I should be capable of recommending for publication.

- Regarding the Sb diffusion, the authors should clarify which core levels they used for Te EDS maps. Can the authors use the Sb-Lb2 to quantify Sb and the Te-La1 for Te to reduce the overlap?
- Can the authors show the dependence of J_z RKKY on the spin-spin interaction J_z . Clarifying the [10,100] meV is a relevant energy scale would be helpful
- Do the authors have any idea about the level of disorder in the films with increasing Sb? The Dirac cone in the $x=0.6-0.7$ looks to have some renormalizations [npj Quantum Materials volume 3, Article number: 29 (2018)] and related papers. I wonder if the observed dip in T_c could be related to disorder. The authors could supplement with afm or tem at different compositions
- Remove the guides to eye in top panel of figure 2

Reviewer #2 (Remarks to the Author):

I thank the authors for their detailed reply to the comments of the referees.

As I stated in my first report, I think this is a very good work. The samples and most of the data is of very high quality and the paper is well written. But even after reading the authors reply I think they are over-interpreting their data. I still believe that they did not prove that the correlation they found could prove a "Dirac Fermion assisted superconductivity".

The authors complain that I did not provide them with any specific details or explanation to support my conclusion. I think it is their role to convince me with their explanation and not the opposite.

Let me summarize the main achievement of the paper and then raise a few concerns that led me to question the authors claim.

In the paper it is shown using ARPES that the Dirac point of the top surface state is at the chemical potential for a certain ratio of Bi/Sb. Using resistance measurements, that essentially measure the entire bi-layer and not only the top surface, the authors show that the minimal T_c happens at the same Bi/Sb ratio. No evidence for SC on the top surface is provided.

In this system SC was observed even for TI layers as thin as 3 QLs where the surface states are expected to be gaped (Nat. Comm. 5, 4247 (2014)). This suggests that the SC does not depend of Dirac fermions.

I understand that the Hall resistance is expected to be dominated by the FeTe layer. Nevertheless, I think it would be a good idea to show it in the paper. If the authors interpretation of the data is correct I expect to see no correlation between the Bi/Sb ratio and the Hall resistance of the bilayer. Do the authors find a Hall resistance that is completely x independent?

ARPES is a surface probe, the fact that for the samples shown in the paper ARPES shows the chemical potential to be in the TI's gap does not mean that the carrier density of the bulk is not changing with the Bi/Sb ratio. This issue becomes even more relevant when taking into account the difference in chemical potential of the two surface states.

In Zhang et al. Nat. Comm. 2, 574 (2011) the Dirac point is at the chemical potential for $x=0.94$

while here the crossing happens at $x=0.8$. This is not a small difference, and it suggests a large charge transfer between the TI and the FeTe layers.

There is at least one report of SC in a layer of FeTe on Bi₂Te₃ (Manna et al. Nat. Comm. 8, 14074 (2017)). In this paper a SC on the FeTe surface gap is shown using STM. One can not rule out that the change in the density of TI surface state on the interface with the Bi/Sb ratio affects the T_c of FeTe. In that case, for example, the topological nature of the surface states play no role.

To summarize, in my opinion the authors failed to prove the claim they make in the title of the manuscript.

Reviewer #3 (Remarks to the Author):

The authors have addressed my comments. I appreciate their efforts, and recommend the publication of this work.

-----Response to reviewers' comments-----

Reviewer #1

The authors do a thorough job responding to my comments. I have a few follow up points, if these can be cleared up, I should be capable of recommending for publication.

We thank Reviewer #1 for his/her new technical comments and recommendation for publication in *Nature Communications*.

Comment 1:

- Regarding the Sb diffusion, the authors should clarify which core levels they used for Te EDS maps. Can the authors use the Sb-L β 2 to quantify Sb and the Te-L α 1 for Te to reduce the overlap?

Fig. R1| EDS spectra of the Sb₂Te₃ and FeTe layers in the 8 QL Sb₂Te₃/50 UC FeTe heterostructure. The Sb and Te peaks in the Sb₂Te₃ layer (top), and the Te peaks in the FeTe layer (bottom). The Sb L _{α 1} peak and the Te L _{α 1} peak in the Sb₂Te₃ layer are shown in purple and green, respectively.

Response: We thank Reviewer #1 for his/her suggestion. For the EDS spectra of the Sb₂Te₃/FeTe heterostructure in Supplementary Fig. 3, we used the Te L _{α 1} peak for Te and the Sb L _{α 1} peak for Sb. We added this information in the caption of Supplementary Fig. 3.

As we noted in our response to **Comment 4** of Reviewer #1 in the first round of review, the Sb L _{β 2} peak overlaps strongly with the Te L _{β 1} peak, and the Te L _{α 1} peak is located near the Sb L _{β 1}

peak (Fig. R1). Therefore, no matter which peak is used to quantify the Sb distribution, the Sb signal will be inevitably influenced by the Te signal in the Sb₂Te₃/FeTe heterostructure. Nevertheless, by narrowing down the selected energy window, we can further minimize the impact of the Te signal on the Sb spectra. We used the new EDS spectra of Sb to replace the old ones in Supplementary Fig. 3 in the revised Supplementary Information.

Comment 2:

- Can the authors show the dependence of J_zRKKY on the spin-spin interaction J_z. Clarifying the [10,100] meV is a relevant energy scale would be helpful.

Response: We apologize to Reviewer #1 for this confusion. In the revised manuscript after the first round of review, we inadvertently used J_z to denote the effective spin-spin interaction J_{eff} . In addition, the z-component J_{RKKY} was labeled as J_z^{RKKY} . Therefore, in the “**Surface Dirac electrons mediated RKKY interaction**” section of Methods, the equation $J_{RKKY} = J_{eff}^2 \times \frac{V_0^2}{AR_{ij}^3} \times \Phi_{zz}(k_F R_{ij})$ on Line 374 demonstrates the dependence of J_{RKKY} on the spin-spin interaction J_{eff} . We rectified all mislabeling errors in the revised manuscript.

The value of the spin-spin interaction J_{eff} in FeTe was estimated to be ~50 meV in prior studies [see Table 1 in Glasbrenner et al. *Nature Phys.* **11**, 953 (2015)]. Therefore, we used a comparable energy scale range [10,100] meV for J_{eff} to estimate the value of the ferromagnetic-type RKKY strength J_{RKKY} in the “**Surface Dirac electrons mediated RKKY interaction**” section of Methods. We cited this paper in the revised manuscript.

Comment 3:

- Do the authors have any idea about the level of disorder in the films with increasing Sb? The Dirac cone in the x=0.6-0.7 looks to have some renormalizations [npj Quantum Materials volume 3, Article number: 29 (2018)] and related papers. I wonder if the observed dip in T_c could be related to disorder. The authors could supplement with afm or tem at different compositions

Response: We thank Reviewer #1 for bringing up this issue. We did carefully think about the disorder effect when we were preparing this manuscript. In the “**Surface Dirac electrons mediated RKKY interaction**” section of Methods, the influence of the disorder on magnetic interaction can be included in an imaginary self-energy broadening $i\Gamma$ in the Green’s function.

The disorder may also give rise to a real part of the self-energy, which might cause “*some renormalizations*” to the Dirac surface states of TI, as mentioned by Reviewer #1. The real part of self-energy can provide corrections to the energy of Dirac points and the velocity of the Dirac surface states, both of which were directly extracted from the experimental data. Therefore, the parameters used in our theory should be treated as “renormalized” values.

Moreover, the disorder in $(\text{Bi}_{1-x}\text{Sb}_x)_2\text{Te}_3$ was found to show a maximum near $x=0.5$, i.e., the compositions of Bi and Sb are equal, through thermal conductivity measurements [Yokota et al. *Jpn. J. Appl. Phys.* **12**, 1205 (1973)]. However, in our experiments, the dip feature in T_c appears near $x=0.85$, which is far from the position for the maximum disorder but close to the Dirac point of the surface states (i.e., $x=0.8$). These observations also rule out the possibility that the appearance of the dip features in both T_c and $\mu_0 H_{c2,\perp}$ is a direct result of the disorder effect in the TI layer.

We followed Reviewer #1’s suggestions and performed STEM measurements on the $x=0.8$ sample near the dip features of both T_c and $\mu_0 H_{c2,\perp}$ (Figs. 4b and 4c of the main text). We observed perfect atomic structures of both the $(\text{Bi}_{0.2}\text{Sb}_{0.8})_2\text{Te}_3$ and FeTe layers and a sharp interface between these two layers (Fig. R2). Compared with the $x=0$ and $x=1$ samples (Fig. 1b, Supplementary Figs. 2 and 4), there is no structure difference in the $x=0.8$ sample. We added the relevant discussion and Fig. R2 in the revised Supplementary Information.

Fig. R2| ADF-STEM image and corresponding EDS maps of the $(\text{Bi}_{0.2}\text{Sb}_{0.8})_2\text{Te}_3/\text{FeTe}$ heterostructure. a,b, The ADF-STEM images of the 8 QL $(\text{Bi}_{0.2}\text{Sb}_{0.8})_2\text{Te}_3/50$ UC FeTe heterostructure. **c-f**, The corresponding EDS maps of Bi, Sb, Fe, and Te of the 8QL $(\text{Bi}_{0.2}\text{Sb}_{0.8})_2\text{Te}_3/50\text{UC}$ FeTe heterostructure.

Comment4:

- Remove the guides to eye in top panel of figure 2

Response: Done.

Reviewer #2

Dear editor, I thank the authors for their detailed reply to the comments of the referees. As I stated in my first report, I think this is a very good work. The samples and most of the data is of very high quality and the paper is well written. But even after reading the authors reply I think they are over-interpreting their data. I still believe that they did not prove that the correlation they found could prove a “Dirac Fermion assisted superconductivity”.

The authors complain that I did not provide them with any specific details or explanation to support my conclusion. I think it is their role to convince me with their explanation and not the opposite.

We thank Reviewer #2 for his/her positive assessment of our work and thoughtful comments in both rounds of review. After reading through Reviewer #2’s new technical comments below, it appears that we have taken for granted (which Reviewer #2 may not be familiar with) the presence of massive Dirac fermions in TI thin films with a hybridization gap between the top and bottom surface states [Chang, Liu, and MacDonald, *Rev. Mod. Phys.* **95**, 011002(2023); Zhang et al. *Nature Phys.* **6**, 584 (2010); Li et al. *Adv. Mater.* **22**, 4002 (2010); Jiang et al. *Phys. Rev. Lett.* **108**, 016401(2012)]. It appears Reviewer #2 thinks that Dirac fermions have to be massless for our mechanism to work, this is not true. As noted in our response to **Comment 2** of Reviwer#2 below, our scenario remains valid when the Dirac fermions have a gap (massive Dirac fermions). For thick 3D TI films with Dirac surface states, the Dirac fermions become massless. Therefore, the terminology “Dirac fermions”, either massive or massless, remains consistent regardless of the TI thickness. For the benefit of readers who may share the same concerns as Reviewer #2, we added a brief discussion on “Dirac fermions” in the revised Supplementary Information.

Comment 1:

Let me summarize the main achievement of the paper and then raise a few concerns that led me to question the authors claim.

In the paper it is shown using ARPES that the Dirac point of the top surface state is at the

chemical potential for a certain ratio of Bi/Sb. Using resistance measurements, that essentially measure the entire bi-layer and not only the top surface, the authors show that the minimal T_c happens at the same Bi/Sb ratio. No evidence for SC on the top surface is provided.

Response: We thank Reviewer #2 for bringing up this issue. As noted in our manuscript, for MBE-grown $(\text{Bi,Sb})_2\text{Te}_3/\text{FeTe}$ heterostructures, the interface-induced superconductivity emerges in the FeTe layer. Our work focuses on exploring the properties of the interfacial superconductivity in $(\text{Bi,Sb})_2\text{Te}_3/\text{FeTe}$ heterostructures rather than the proximity-induced superconductivity on the top surface of the TI layer. Besides the transport properties of the interfacial superconductivity in $(\text{Bi,Sb})_2\text{Te}_3/\text{FeTe}$ heterostructures, what we need is the chemical potential of the TI layer near the $(\text{Bi,Sb})_2\text{Te}_3/\text{FeTe}$ interface. In the first round of review, we have explained why we used the ARPES measurements rather than the Hall measurements to determine the chemical potential of the $(\text{Bi,Sb})_2\text{Te}_3$ layer near the $(\text{Bi,Sb})_2\text{Te}_3/\text{FeTe}$ interface. Please see the details in our responses to **Comment 3** of **Reviewer #1** and **Comment 1** of **Reviewer #2** in the first round of review.

As noted in our response to **Comment 5** of **Reviewer #2** in the first round of review, we agreed with Reviewer #2 that the charge transfer between $(\text{Bi}_{1-x}\text{Sb}_x)_2\text{Te}_3$ and FeTe can lead to the Dirac points on the top and bottom [i.e. the $(\text{Bi}_{1-x}\text{Sb}_x)_2\text{Te}_3/\text{FeTe}$ interface] surfaces of $(\text{Bi}_{1-x}\text{Sb}_x)_2\text{Te}_3$ at different energies. Because of this energy difference, the minimum value of T_c should be observed away from the Bi/Sb ratio at which the chemical potential crosses the Dirac point on the top surface of $(\text{Bi}_{1-x}\text{Sb}_x)_2\text{Te}_3$. In our experiments, the chemical potential crosses the Dirac point at $x=0.8$ (Fig. 4a of the main text). However, the minimum values of both $T_{c, \text{onset}}$ and $T_{c,0}$ were observed at $x=0.85$ (Fig. 4b of the main text). This observation is consistent with the occurrence of the charge transfer between $(\text{Bi}_{1-x}\text{Sb}_x)_2\text{Te}_3$ and FeTe, i.e., the Dirac points on the top and bottom surfaces of the $(\text{Bi}_{1-x}\text{Sb}_x)_2\text{Te}_3$ layer are located at slightly different energies.

Finally, we would like to discuss the proximity-induced superconductivity on the top surface of the TI layer. The proximity-induced superconductivity on the top surface of the TI layer greatly relies on the thickness and carrier density (i.e., the chemical potential) of the TI layer [Stellhorn, *Interplay of Proximity Effects in Superconductor/ferromagnet Heterostructures* (Forschungszentrum Jülich GmbH, Zentralbibliothek, Verlag, 2021); Buzdin, *Rev. Mod. Phys.* **77**, 935(2005)]. We agree with Reviewer #2 that a systematic investigation of the proximity-

induced superconducting gap size on the top surface of the MBE-grown $(\text{Bi}_{1-x}\text{Sb}_x)_2\text{Te}_3/\text{FeTe}$ heterostructures as a function of the Sb concentration and the thickness of the $(\text{Bi}_{1-x}\text{Sb}_x)_2\text{Te}_3$ layers is an interesting project. As noted above, this topic is beyond the scope of the current work. We note that prior scanning tunneling microscopy/spectroscopy (STM/S) measurements have shown the proximity effect-induced superconducting gap on the top surface of a 5 QL $\text{Bi}_2\text{Te}_3/\text{FeTe}$ heterostructure [Qin et al. *Nano Lett.* **20**, 3160 (2020)].

Comment 2:

In this system SC was observed even for TI layers as thin as 3 QLs where the surface states are expected to be gaped (Nat. Comm. 5, 4247 (2014)). This suggests that the SC does not depend of Dirac fermions.

Response: Reviewer #2 is correct that the interface-induced superconductivity emerges in MBE-grown $(\text{Bi,Sb})_2\text{Te}_3/\text{FeTe}$ heterostructures with the TI layer as thin as 3 QL or even thinner. For TI thin films in the 2D regime, a hybridization gap between the top and bottom surface states is formed [Chang, Liu, and MacDonald, *Rev. Mod. Phys.* **95**, 011002(2023); Zhang et al. *Nature Phys.* **6**, 584 (2010); Li et al. *Adv. Mater.* **22**, 4002 (2010); Jiang et al. *Phys. Rev. Lett.* **108**, 016401(2012)]. However, as noted above, the appearance of the hybridization gap in TI thin films leads to the Dirac fermions acquiring mass (or a gap) and thus makes Dirac fermions massive [Liu et al. *Phys. Rev. B* **81**, 041307 (2010); Lv et al. *Phys. Rev. B* **81**, 115407 (2010)]. In other words, the terminology of “Dirac fermions” remains consistent regardless of the TI thickness. We note that our theoretical analysis remains applicable to both massless and massive Dirac fermions. In Supplementary Fig. 19, we have demonstrated that both intra-band scattering and inter-band scattering give rise to the RKKY interaction. In particular, when the chemical potential is close to the Dirac point, the intra-band contribution is negligible due to the vanishing density of states at the Dirac point while the inter-band contribution is the dominant mechanism (Fig. R3). In the case of massive Dirac fermions, the inter-band contribution plays a primary role in RKKY interaction and is of ferromagnetic type when the chemical potential is inside the Dirac fermion gap. This inter-band contribution is also known as the van Vleck mechanism [van Vleck, *The Theory of Electric and Magnetic Susceptibilities* (Oxford) (1965)] or the Bloembergen-Rowland mechanism [Bloembergen and Rowland, *Phys. Rev.* **97**, 1679(1955)], which was previously known to play an important role in inducing ferromagnetism in magnetically doped TI films [Yu et al. *Science* **329**, 61(2010)]. Please see our responses to **Comments 2** and **3** of

Reviewer #3 in the first round of review. We added a brief discussion on “Dirac fermions” and added Fig. R3 as Supplementary Fig. 19e in the revised Supplementary Information.

Fig. R3 | RKKY interaction $\Phi_{zz}(z)$ as a function of z at disorder broadening $\Gamma = 0.6$. Both intra-band and inter-band scatterings contribute to the total RKKY interaction. As k_F approaches 0, the contribution from intra-band contribution vanishes.

Comment 3:

I understand that the Hall resistance is expected to be dominated by the FeTe layer. Nevertheless, I think it would be a good idea to show it in the paper. If the authors interpretation of the data is correct I expect to see no correlation between the Bi/Sb ratio and the Hall resistance of the bilayer. Do the authors find a Hall resistance that is completely x independent?

Response: As noted in our responses to **Comment 3** of **Reviewer #1** and **Comment 1** of **Reviewer #2** in the first round of review, FeTe is an antiferromagnetic metal, which has higher conductivity as compared to the $(\text{Bi}_{1-x}\text{Sb}_x)_2\text{Te}_3$ layer. Therefore, the Hall trace of the $(\text{Bi}_{1-x}\text{Sb}_x)_2\text{Te}_3/\text{FeTe}$ heterostructure does not accurately reflect the carrier density or the position of the chemical potential in the $(\text{Bi}_{1-x}\text{Sb}_x)_2\text{Te}_3$ layer. **Figure R4** shows the Hall traces of all 8 QL $(\text{Bi}_{1-x}\text{Sb}_x)_2\text{Te}_3/50\text{UC FeTe}$ heterostructures with interfacial superconductivity and the 50 UC FeTe layer without superconductivity. We found that the absolute values of the Hall trace slopes are small for all these samples. For $0 \leq x \leq 0.6$, the Hall traces of these $(\text{Bi}_{1-x}\text{Sb}_x)_2\text{Te}_3/\text{FeTe}$ heterostructures show systematic behavior, and the Hall trace slope decreases with increasing x . However, for x near 0.85, i.e., $0.7 \leq x \leq 1.0$ and the chemical potential is near the Dirac point of the TI layer, the Hall traces show some fluctuations.

Next, we employed a two-layer structure to model the TI/FeTe heterostructure and performed simple calculations to understand our observations. To simplify our calculations, we ignored the charge transfer effect between TI and FeTe layers. The total Hall resistance of the TI/FeTe bilayer can be calculated using the following equation:

$$R_{yx} = \left(\frac{\frac{R_{yx}^{FeTe}}{(R_{xx}^{FeTe})^2} + \frac{R_{yx}^{TI}}{(R_{xx}^{TI})^2}}{\left(\frac{1}{R_{xx}^{FeTe}} + \frac{1}{R_{xx}^{TI}} \right)^2} \right) \quad (1)$$

As noted above, the FeTe layer is an antiferromagnetic metal, which exhibits higher conductivity compared to the $(\text{Bi}_{1-x}\text{Sb}_x)_2\text{Te}_3$ layer, so R_{xx}^{FeTe} is much less than R_{xx}^{TI} (i.e., $R_{xx}^{FeTe} \ll R_{xx}^{TI}$).

Fig. R4| Hall traces of all 8 QL $(\text{Bi}_{1-x}\text{Sb}_x)_2\text{Te}_3/50\text{UC FeTe}$ heterostructures with interfacial superconductivity and 50 UC FeTe layer without superconductivity. All measurements are taken at $T=30$ K.

Given this assumption, we can simplify Equation (1):

$$R_{yx} = \left(R_{yx}^{FeTe} + R_{yx}^{TI} * \left(\frac{R_{xx}^{FeTe}}{R_{xx}^{TI}} \right)^2 \right) = \left(R_{yx}^{FeTe} + \frac{R_{yx}^{TI}}{(R_{xx}^{TI})^2} * (R_{xx}^{FeTe})^2 \right) \quad (2)$$

Therefore, the total Hall resistance R_{yx} value of the TI/FeTe heterostructure is determined by the joint contribution of the Hall resistance of the FeTe layer R_{yx}^{FeTe} and the modified Hall resistance of the TI layer $R_{yx}^{TI} * \left(\frac{R_{xx}^{FeTe}}{R_{xx}^{TI}} \right)^2$. For all $(\text{Bi}_{1-x}\text{Sb}_x)_2\text{Te}_3/\text{FeTe}$ heterostructures, both R_{xx}^{FeTe} and R_{yx}^{FeTe} are assumed to be constant. Systematic change in x can affect both R_{xx}^{TI}

and R_{yx}^{TI} . For $0 \leq x \leq 0.6$, the value of $\frac{R_{yx}^{TI}}{(R_{xx}^{TI})^2}$ remains relatively insensitive to x and thus results in systematic behaviors in R_{yx} of the TI/FeTe heterostructures. However, for $0.7 \leq x \leq 1.0$, specifically when the chemical potential is near the Dirac point of the TI layer, the value of $\frac{R_{yx}^{TI}}{(R_{xx}^{TI})^2}$ becomes more sensitive to x and thus leads to fluctuations in the Hall traces. We note that the carrier type of the FeTe layer is n -type (Figs. R4 and Supplementary Fig. S16), a change in the Hall trace sign of the $(\text{Bi}_{1-x}\text{Sb}_x)_2\text{Te}_3/\text{FeTe}$ heterostructures is in the range of $0.6 < x < 0.7$, as opposed to occurring beyond $x=0.8$, where the chemical potential crosses the Dirac point based on ARPES (Figs. 2 and 4a). This observation also suggests that there is no substantial charge transfer effect between the $(\text{Bi}_{1-x}\text{Sb}_x)_2\text{Te}_3$ layer and FeTe layers.

We added Fig. R4 and relevant discussion in the revised Supplementary Information.

Comment 4:

ARPES is a surface probe, the fact that for the samples shown in the paper ARPES shows the chemical potential to be in the TI's gap does not mean that the carrier density of the bulk is not changing with the Bi/Sb ratio. This issue becomes even more relevant when taking into account the difference in chemical potential of the two surface states.

Response: Reviewer #2 is correct that “*ARPES is a surface probe*”. As noted in our responses to **Comment 3** of **Reviewer #1** and **Comment 1** of **Reviewer #2** in the first round of review, our prior work [Zhang et al. *Nature Commun.* **2**, 574 (2011)] has demonstrated that for a series of MBE-grown $(\text{Bi}_{1-x}\text{Sb}_x)_2\text{Te}_3$ layers, the k_F value obtained from ARPES measurements agrees well with the carrier density achieved based on Hall transport measurements.

As noted in our response to **Comment 1** of **Reviewer #2** above, we agree with Reviewer #2 that the charge transfer between $(\text{Bi}_{1-x}\text{Sb}_x)_2\text{Te}_3$ and FeTe can lead to the Dirac points on the top and bottom [i.e. the $(\text{Bi}_{1-x}\text{Sb}_x)_2\text{Te}_3/\text{FeTe}$ interface] surfaces of $(\text{Bi}_{1-x}\text{Sb}_x)_2\text{Te}_3$ at different energies. A large energy difference may affect the claim of our work. However, this energy difference is small based on a prior ARPES work on the $\text{Bi}_2\text{Te}_3/\text{FeTe}$ heterostructure. The chemical potential difference is estimated to be ~ 10 meV by comparing the band structures of the 2QL $\text{Bi}_2\text{Te}_3/\text{FeTe}$ heterostructure and the FeTe layer [Fig. 3g in Owada et al. *Phys. Rev. B* **100**, 064518 (2019)]. Moreover, the Hall trace fluctuations for the $0.7 \leq x \leq 1.0$ samples also imply that R_{yx}^{TI} is large and the TI layer is insulating. Please see our discussion in our response to **Comment 3** of

Reviewer #2 above.

Comment 5:

In Zhang et al. Nat. Comm. 2, 574 (2011) the Dirac point is at the chemical potential for $x=0.94$ while here the crossing happens at $x=0.8$. This is not a small difference, and it suggests a large charge transfer between the TI and the FeTe layers.

Response: We thank Reviewer #2 for bringing up this issue. The value of the nominal Sb concentration x is usually different for the $(\text{Bi}_{1-x}\text{Sb}_x)_2\text{Te}_3$ layers grown by different groups using different MBE chambers. For the $(\text{Bi}_{1-x}\text{Sb}_x)_2\text{Te}_3$ layer with the lowest carrier density, i.e. when the chemical potential is crossing its Dirac point, the x value is found to be ~ 0.94 for the Tsinghua group [Zhang et al. *Nature Commun.* **2**, 574 (2011); **Chang** grew these samples when he was a graduate student at Tsinghua], ~ 0.84 for the University of Tokyo group [Yoshimi et al. *Nature Commun.* **6**, 6627 (2015)], ~ 0.47 for the UCLA group [Zou et al. *Appl. Phys. Lett.* **110**, 212401 (2017)], and ~ 0.8 at Penn State in the current work. The variation in the cited value of x for the film with the lowest carrier density from different groups and different MBE chambers could be traced to different (and difficult to calibrate across different MBE chambers) the myriad film growing conditions, such as the temperatures of the different Knudsen tubes, the substrate temperatures, the vacuum of the MBE chamber, etc. From an MBE grower point of view, it is not scientifically meaningful to compare the two x values for the two $(\text{Bi}_{1-x}\text{Sb}_x)_2\text{Te}_3$ thin film samples grown by two different MBE chambers at separate institutions. It is only meaningful to compare the x value of samples iteratively grown in the same chamber during the same experimental run. We note that the x value difference in the two samples mentioned by Reviewer #2 does not imply the occurrence of the large charge transfer between the TI and FeTe layers, see our response to **Comment 4** of **Reviewer #2** above.

Comment 6:

There is at least one report of SC in a layer of FeTe on Bi_2Te_3 (Manna et al. Nat. Comm. 8, 14074 (2017)). In this paper a SC on the FeTe surface gap is shown using STM. One can not rule out that the change in the density of TI surface state on the interface with the Bi/Sb ratio affects the T_c of FeTe. In that case, for example, the topological nature of the surface states play no role.

Response: We thank Reviewer #2 for bringing up this reference. We did know this paper well. Before we directly address this comment, we first would like to bring the synthesis method used

in this *Nature Communications* work to Reviewer#2's attention. The FeTe/Bi₂Te₃ heterostructures are synthesized by directly depositing 0.5~1 monolayer Fe atoms on top of Bi₂Te₃ bulk crystals at $T = 300$ K and then annealing the sample at $T = 315$ °C for 15 min. The top Fe atoms react with Bi₂Te₃ bulk crystals during the annealing process, i.e. Fe atoms selectively attract Te atoms from the Bi₂Te₃ compound to form the top "FeTe" layer [see Methods section of Manna et al. *Nature Commun.* **8**, 14074 (2017)]. With such a process, it is reasonable to expect a high density of Te vacancies in Bi₂Te₃ near the FeTe/Bi₂Te₃ interface. Therefore, it would not be accurate to assume that the FeTe/Bi₂Te₃ heterostructures used in *Nature Communications* work share the same properties as the MBE-grown Bi₂Te₃/FeTe used in our work.

We are puzzled by Reviewer #2's comments "*One can not rule out that the change in the density of TI surface state on the interface with the Bi/Sb ratio affects the T_c of FeTe*" and "*In that case, for example, the topological nature of the surface states play no role.*". It appears to us that there is an internal inconsistency in these two statements. As noted in our manuscript, the interfacial superconductivity emerges in the FeTe layer when it comes into contact with the TI layer. Therefore, in this *Nature Communications* work, even under the assumption of ideal FeTe/Bi₂Te₃ heterostructures, it remains a reasonable expectation to detect a superconducting gap on the top surface of the FeTe layer through STM/S measurements [Manna et al. *Nature Commun.* **8**, 14074 (2017)].

Finally, we would like to bring the first sentence of the last paragraph in this *Nature Communications* work "*We finally note that the leakage of the gap into the TI substrate across the FeTe–Bi₂Te₃ interface (Fig. 3c) indicates the presence of superconducting correlations in the TI material close to the interface.*" to Reviewer#2's attention. This statement agrees well with the claim of our work.

Comment 7:

To summarize, in my opinion the authors failed to prove the claim they make in the title of the manuscript.

Response: We hope our responses together with the changes in the revised manuscript convinced Reviewer #2 that we have indeed justified the claim of our paper.

Reviewer #3

The authors have addressed my comments. I appreciate thier efforts, and recommend the publication of this work.

We thank Reviewer #3 for his/her recommendation for publication in *Nature Communications*.

-----List of changes-----

(All the changes in the main article are shown in blue)

1. Line 31 on Page 2, we rewrote the below sentence.

“We **provide evidence to show that the observed** interfacial superconductivity and its **chemical potential dependence is the result of the competition between the Ruderman-Kittel-Kasuya-Yosida-type ferromagnetic coupling mediated by Dirac surface states and antiferromagnetic exchange couplings that generate the** bicollinear antiferromagnetic order in the FeTe layer.”

2. Line 71 on Page 4, we rewrote the below sentence.

“However, **there remains an important obstacle, specifically,** the superconductivity in MBE-grown thin superconducting films usually disappears once a TI layer is grown on top, presumably due to the occurrence of charge transfer ¹⁹.”

3. Line 382 on Page 18, we added the below sentence.

“**The real part of self-energy provides corrections to the energy of Dirac points and the velocity of the Dirac surface states** ⁵².”

4. Line 401 on Page 18, we rewrote the below sentence.

“**When k_F is near the Dirac point, the inter-band contribution dominates, and a larger RKKY interaction J_{RKKY} is achieved (Supplementary Fig. 19).**”

5. Line 407 on Page 19, we rewrote the below sentence.

“**In FeTe, the value of J_{eff} has been determined to be ~ 50 meV (Ref.⁵³). Therefore, we assume a comparable energy scale range of [10,100] meV for J_{eff} and estimate the ferromagnetic-type RKKY strength J_{RKKY} within the range of $-[0.02, 1.9]$ meV at $k_F \rightarrow 0$.**”

6. We removed the guide dashed lines in the top panel of Fig. 2 in the revised manuscript.

7. We added **Sections II.1 to 3** in the revised Supplementary Information.

8. We added **Supplementary Figs. 3, 17, and 19e** in the revised Supplementary Information.

9. We added two references shown in blue in the revised manuscript.

10. We made numbers of minor revisions shown in blue in the revised manuscript and Supplementary Information.

REVIEWERS' COMMENTS

Reviewer #2 (Remarks to the Author):

I thank the authors for the effort they made to answer my questions.

-----Response to reviewers' comments-----

Reviewer #2

I thank the authors for the effort they made to answer my questions.

We thank Reviewer #2 for his/her recommendation for publication in *Nature Communications*.